# A deep learning approach to programmable RNA switches

Nicolaas M. Angenent-Mari[1,2,3,8], Alexander S. Garruss [3,4,5,8], Luis R. Soenksen [1,2,3,6,8], George Church[3,5,7] & James J. Collins [1,2,3,6,7 ✉]

Engineered RNA elements are programmable tools capable of detecting small molecules, proteins, and nucleic acids. Predicting the behavior of these synthetic biology components remains a challenge, a situation that could be addressed through enhanced pattern recognition from deep learning. Here, we investigate Deep Neural Networks (DNN) to predict toehold switch function as a canonical riboswitch model in synthetic biology. To facilitate DNN training, we synthesize and characterize in vivo a dataset of 91,534 toehold switches spanning 23 viral genomes and 906 human transcription factors. DNNs trained on nucleotide sequences outperform ($R^2 = 0.43$–0.70) previous state-of-the-art thermodynamic and kinetic models ($R^2 = 0.04$–0.15) and allow for human-understandable attention-visualizations (VIS4Map) to identify success and failure modes. This work shows that deep learning approaches can be used for functionality predictions and insight generation in RNA synthetic biology.

[1] Department of Biological Engineering, Massachusetts Institute of Technology (MIT), Cambridge, MA 02139, USA. [2] Institute for Medical Engineering and Science (IMES), MIT, Cambridge, MA 02139, USA. [3] Wyss Institute for Biologically Inspired Engineering, Harvard University, Boston, MA 02115, USA. [4] Program in Bioinformatics and Integrative Genomics, Harvard University, Cambridge, MA 02138, USA. [5] Department of Genetics, Harvard Medical School, Boston, MA 02115, USA. [6] Department of Mechanical Engineering, MIT, Cambridge, MA 02139, USA. [7] Harvard-MIT Program in Health Sciences and Technology, Cambridge, MA 02139, USA. [8] These authors contributed equally: Nicolaas M. Angenent-Mari, Alexander S. Garruss, Luis R. Soenksen. ✉email: jimjc@mit.edu

Engineered ribonucleic acid (RNA) molecules with targeted biological functions play an important role in synthetic biology[1], particularly as programmable response elements for small molecules, proteins, and nucleic acids. Examples include riboswitches, riboregulators, and ribozymes, many of which hold great promise for a variety of in vitro and in vivo applications[1,2]. Despite their appeal, the design and validation of this emerging class of synthetic biology modules have proven challenging due to variability in function that remains difficult to predict[2–9]. Current efforts aiming to unveil fundamental relationships between RNA sequence, structure, and behavior focus mostly on mechanistic thermodynamic modeling and low-throughput experimentation, which often fail to deliver sufficiently predictive and actionable information to aid in the design of complex RNA tools[2–9]. Deep learning, by contrast, constitutes a set of computational techniques well suited for feature recognition in complex and highly combinatorial biological problems[10–14], such as the sequence design space of synthetic RNA tools. However, the application of deep learning to predicting function in RNA synthetic biology has been limited by a notable scarcity of datasets large enough to effectively train deep neural networks (DNN). Toehold switches, in particular, represent a benchmark RNA element in synthetic biology that could greatly benefit from deep-learning approaches to better predict function and elucidate useful design rules.

Toehold switches are a class of versatile prokaryotic riboregulators inducible by the presence of a fully programmable trans-RNA trigger sequence[2–6,15,16]. These RNA synthetic biology modules have displayed impressive dynamic range and orthogonality when used both in vivo as genetic circuit components[2,5,6], and in vitro as nucleic acid diagnostic tools utilizing cell-free protein synthesis (CFPS) systems[3,4,15,16]. Similar to other RNA synthetic biology tools, a substantial fraction of toehold switches show poor to no measurable function when tested experimentally, and while efforts have been made to establish rational, mechanistic rules for improved performance based on low-throughput datasets[2–9,15,16], the practical utility of these approaches remains inconclusive. Thus, considering the wide applicability and general challenges of toehold-switch design, our objective in this study is to develop a deep-learning platform to predict toehold-switch function as a canonical RNA switch model in synthetic biology.

To achieve our goal in collaboration with Valeri et al.[17], we first expand the size of available toehold datasets using a high-throughput DNA synthesis and sequencing pipeline to characterize over $10^5$ toehold switches. We then use this comprehensive dataset to demonstrate that deep neural networks trained directly on switch RNA sequences can outperform rational thermodynamic and kinetic analyses to predict toehold-switch function. Furthermore, we enhance the transparency of our deep-learning approach by utilizing a nucleotide complementarity matrix input representation to visualize important learned secondary-structure patterns in selected models. This attention-visualization technique, which we term VIS4Map (Visualizing Secondary Structure Saliency Maps), allows us to identify RNA module success and failure modes by discovering secondary structures that our deep-learning model uses to accurately predict toehold-switch function. The resulting dataset, models, and visualization analysis (Fig. 1) represent a substantial step forward for the validation and interpretability of high-throughput approaches to designing RNA synthetic biology tools, surpassing the limits of current mechanistic RNA secondary-structure modeling.

## Results

**Library synthesis and validation.** A fundamental hurdle in applying deep-learning techniques to RNA synthetic biology systems is the limited size of currently published datasets, which are notably smaller than typical dataset sizes required for the training of deep network architectures in other fields[10,18–21]. For example, to date, <1000 total toehold switches have been designed and tested[2–6,9,15,16]. While a recent attempt was made to apply deep learning to a riboswitch dataset with 263 variants[22], the lack of high-throughput datasets has generally limited the synthetic biology community's ability to analyze this type of response molecule using deep-learning techniques. High-throughput assays that utilize deep sequencing to analyze fluorescence-sorted bacteria have previously been used to characterize the translation of *Escherichia coli* mRNA[23–27]; in this study, in order to improve our understanding and ability to predict new functional RNA-based response elements, we synthesized and characterized an extensive in vivo library of toehold switches using a high-throughput flow-seq (also known as sort-seq)[23,24] pipeline for subsequent exploration using various machine-learning and deep-learning architectures.

Our toehold-switch library was designed and synthesized based on a large collection (244,000) of putative trigger sequences, spanning the complete genomes of 23 pathogenic viruses, the entire coding regions of 906 human transcription factors, and ~10,000 random sequences. From a synthesized oligo pool, we generated two construct libraries, for ON and OFF states, which were subsequently transformed into BL21 *E. coli* (Fig. 1 and Supplementary Fig. 1a–e). The first library contained OFF toehold-switch constructs that lacked a trigger, while the second library of ON constructs contained the same toeholds with complementary triggers fused to their corresponding switches. The two libraries were then sorted on a fluorescence-activated cell sorter (FACS) using four bins (Fig. 1 and Supplementary Figs. 1d, e, 2a), and the toehold-switch variants contained in each bin were quantified using next-generation sequencing (NGS) to recover their individual fluorescence distributions from raw read counts (Fig. 1). After quality control (Supplementary Table 1), the toehold-switch library contained 109,067 ON-state measurements (Fig. 2a), 163,967 OFF-state measurements (Fig. 2b), and 91,534 ON/OFF paired ratios (Fig. 2c), where both ON and OFF states were characterized for a given switch (Fig. 2e, f). ON and OFF data were normalized from 0 to 1, resulting in an ON/OFF ratio normalized from −1 to 1 (see Supplementary methods). Both ON and OFF data spanned the full range of measured GFP signals, meaning that some ON switches failed to induce and expressed no measurable GFP signal, while some OFF switches failed to repress ribosome binding and leaked the maximum measurable GFP signal. Additionally, it should be noted that while ON data are relatively uniform in distribution, OFF data are highly skewed towards low-signal variants (see Supplementary methods section for a detailed discussion of data balancing).

Since RNA synthetic biology tools such as toehold switches are often used within in vitro cell-free systems[3,4,15,16], we validated our in vivo ON/OFF measurements in an in vitro setting to ensure they were reasonable indicators of switch performance in a CFPS system. To achieve this, we selected eight high-performance switches and eight low-performance switches, and individually cloned and characterized each one in a PURExpress CFPS (Fig. 1, Supplementary Fig. 5, and Supplementary Table 2). All low-performance switches showed no induction, while the high-performance switches showed a spread of ON/OFF ratios between 2 and 13 ($P < 0.0001$ between high and low switches, two-tailed $t$ test). The wide range of GFP expression seen from the high-performance switches results from a relatively weak rank-order correlation we have observed between the performance of our toeholds in vivo and in vitro (Supplementary Fig. 1c), which differs from other work comparing RNA actuators in living cells and cell-free systems[28]. The effect may stem from

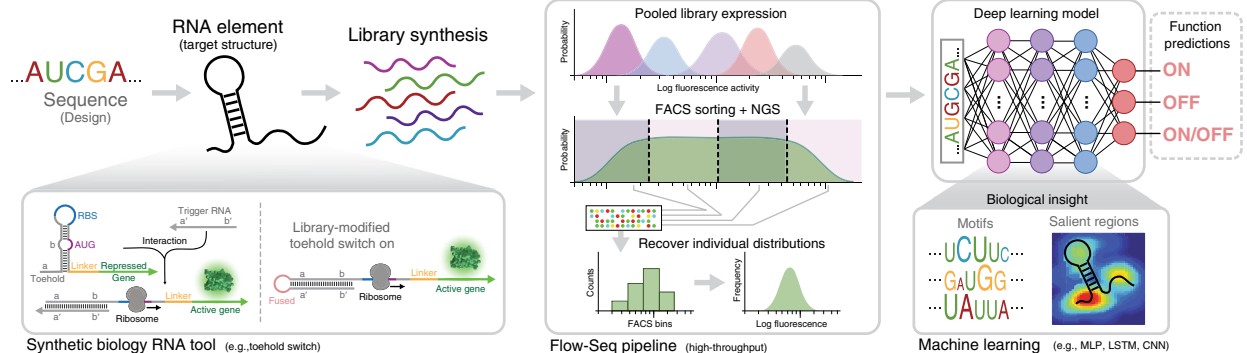

**Fig. 1 Deep learning for ribonucleic acid (RNA) synthetic biology pipeline.** RNA tool selection is followed by library synthesis and characterization with analysis using deep neural networks (DNN) to provide functionality predictions and biological insights. We used a high-throughput toehold-switch library as a canonical model for the general investigation of RNA synthetic biology tools. The original toehold-switch architecture from Green et al.[2] was used, containing a 12-nucleotide toehold (a/a′) and an 18-nucleotide stem (b/b′) fully unwound by the trigger (left-bottom). We selected to fuse the RNA trigger to the 5′ end of the switch by an unstructured linker to facilitate library synthesis. A flow-sequence (seq) pipeline was used to characterize the fluorescence signal of individual toehold switches in a pooled sequential assay, including pooled induction, fluorescence-activated cell sorter (FACS) sorting, next-generation sequencing (NGS), and count frequency analysis. Finally, various DNN architectures were used to predict data outputs, while features contributing to DNN predictions were intuitively visualized to elucidate biological insights. Center panel adapted from Peterman et al.[24].

differences in trigger-toehold interactions between the in vivo cellular environment and the in vitro cell-free environment. Nonetheless, these results indicate that while the performance of toehold switches in vivo and in vitro may differ, in vivo measurements can still be used to classify categorically whether a switch will function in vitro.

**Rational analysis using RNA secondary-structure models.** Before initiating the exploration of deep-learning models to predict function in our large-scale toehold-switch library, we sought to determine whether traditional tools for analyzing synthetic RNA modules could be used to accurately predict toehold-switch behavior, including k-mer searches and mechanistic modeling utilizing thermodynamic and kinetic parameters. K-mer searches of biological sequence data are often used to discover motifs, and while certain overrepresented motifs were found in our dataset (Fig. 3a and Supplementary Table 3), utilization of these did not significantly improve functional predictions of switch behavior. Other current state-of-the-art approaches for designing RNA synthetic biology tools primarily analyze secondary structure using thermodynamic principles[29–31]. Following such prior works, we used NUPACK[29] and ViennaRNA[31] software packages to calculate a total of 30 rational features for our entire library, including the minimum free energy (MFE), ideal ensemble defect (IED), and native ensemble defect (NED) of the entire toehold-switch library as well as various sub-segments in each sequence (Supplementary Table 4). A number of these parameters had previously been reported to correlate with experimental toehold-switch ON/OFF measurements for smaller datasets[2], and NUPACK's design algorithm, in particular, is set to optimize IED when proposing target RNA secondary structures[3,4,15,29]. However, when analyzing these rational features with our larger dataset, we found them to be poor predictors of toehold-switch function (Fig. 3b and Supplementary Fig. 6). In modest agreement with the findings of Green et al.[2], the MFE of the RBS-linker region showed the highest correlation of this feature set for ON/OFF ($R^2$: ON = 0.14, OFF = 0.06, ON/OFF = 0.04), with NUPACK's IED also showing above-average correlation ($R^2$: ON = 0.07, OFF = 0.02, ON/OFF = 0.03). While measurable, these correlation metrics were too weak for practical use in computer-aided design of this specific RNA synthetic biology tool[3,4,15,29].

We next explored the use of more complex thermodynamic models that take into account well-established hypotheses for translation initiation and the ribosome docking mechanism in combination with multiple thermodynamic features to improve their predictions[32–37]. One of the most developed of these models is the ribosome-binding site (RBS) calculator (v2.1; Salis Lab), which is a comprehensive regression model parameterized on thousands of curated RBS variants[32–35]. We used the RBS calculator to predict the ON and OFF translation initiation rates for our toehold switches, but also found low predictive performance comparable to other rational features (Fig. 3b) when tested on our database ($R^2$: ON = 0.09, OFF = 0.05, ON/OFF = 0.0001).

One potential explanation for the limited predictive power of current thermodynamic models for RNA folding tasks concerns the influence of kinetically stable secondary-structure intermediates that may compete with thermodynamic equilibrium states[35,38]. To determine whether a kinetic analysis of toehold-switch folding dynamics could help explain our experimental results, we calculated four additional features based on kinetic trajectories using the Kinfold package[39] (Supplementary Fig. 7). As with predictions obtained using other thermodynamic models, these kinetic features showed poor correlations ($R^2$: ON = 0.04, OFF = 0.04, ON/OFF = 0.001 for the best feature) to our empirical dataset (Supplementary Fig. 7e). Considering these results, the cause of limited functional predictions from thermodynamic and kinetic RNA secondary-structure models remains unclear but may stem from the use of potentially incomplete energetic models, incorrect mechanistic hypotheses, or interference from the in vivo context of the bacterial cell. Regardless of the source of error, we sought to explore deep learning as a machine-learning paradigm to develop models with higher predictive abilities than previously reported, with the hope of allowing useful computer-aided systems for the design of RNA synthetic biology tools.

**Improved prediction using multilayer perceptron models.** Given that simple regression models based on previous state-of-the-art RNA thermodynamic and kinetic calculations were ineffective at predicting toehold-switch performance, we next tested the use of feed-forward neural networks, also known as multilayer perceptron (MLP) models, as a baseline architecture for our investigation (Fig. 3c). We first trained a three-layer MLP model on our dataset with an input consisting of the 30 previously calculated thermodynamic rational features (see "Methods" for

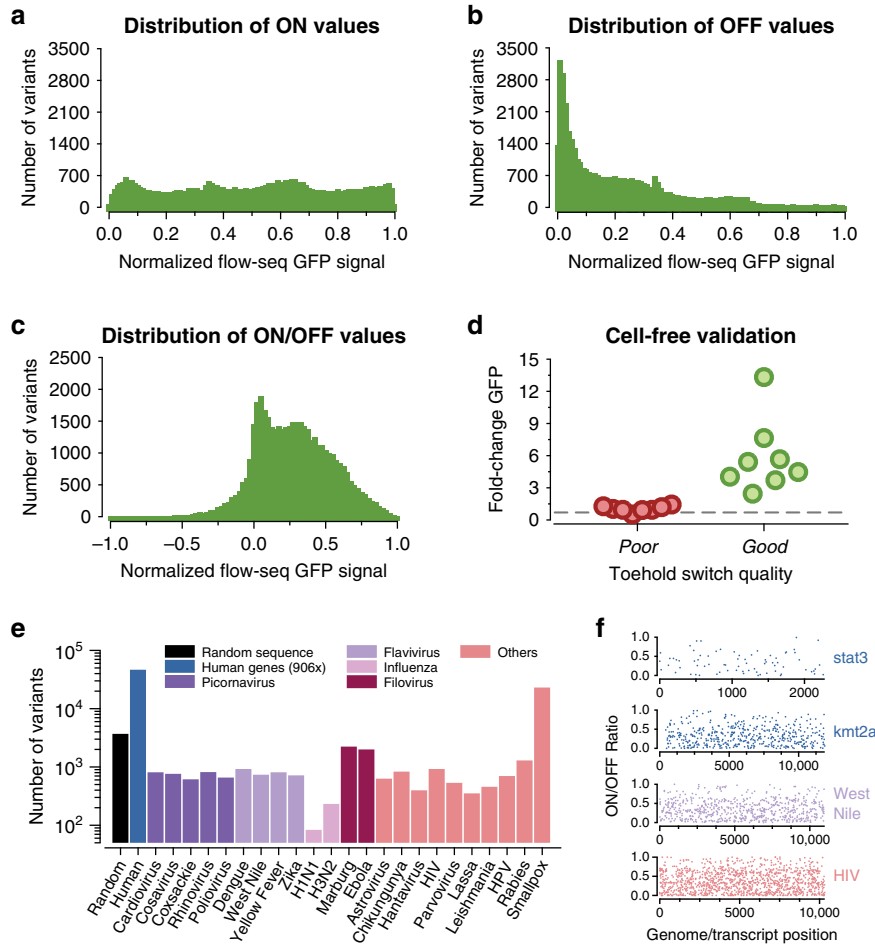

**Fig. 2 Flow-seq toehold-switch library characterization and trigger ontology.** The distribution of recovered toeholds for (**a**) ON-state signals, (**b**) OFF-state signals, and (**c**) calculated ON/OFF ratios are shown. **d** Validation results for toehold switches expressed in a PURExpress cell-free system with un-fused-trigger RNA, including eight low-performing (poor, ON/OFF < 0.05) and eight high-performing (good, ON/OFF > 0.97) samples. Obtained in vivo flow-seq data show competency in classifying switch performance for this in vitro cell-free biological context ($P < 0.0001$ between high and low switches, two-tailed $t$ test) with $n = 3$ biologically independent samples each for both ON and OFF measurements. **e** Tested switch/trigger variants from each origin category, including randomly generated sequences, 906 human transcription factor transcripts, and 23 pathogenic viral genomes. **f** Experimental ON/OFF ratios for all triggers tiled across the transcripts of two clinically relevant human transcription factors (*stat3* and *kmt2a*) upregulated in cancerous phenotypes[51,52], as well as all triggers tiled across the genomes of two pathogenic viruses: West Nile Virus (WNV) and human immunodeficiency virus (HIV). GFP  green fluorescent protein, Seq sequence, HPV  human papillomavirus. All ON, OFF, and ON/OFF values shown were selected from quality control process #3, QC3 in Supplementary Fig. S13 and Supplementary Table 1. All source data are provided as a Source Data file.

further detail). When trained in regression mode, this MLP model was able to deliver better predictions than any of the individual rational features or the RBS calculator based on $R^2$ and mean absolute error (MAE) ($R^2$: ON = 0.35, OFF = 0.25, ON/OFF = 0.20) (Fig. 3d, e). Similarly, when this model was trained in classification mode (ON/OFF: binarized at +/− 0.7, Supplementary Fig. 8), it achieved a 0.76 area under the receiver–operator curve (AUROC) and 0.18 area under the precision-recall curve (AUPRC), as seen in Fig. 3f. The MLP model slightly outperformed a logistic regressor trained on the same rational features (Fig. 3d–f), suggesting that the MLP architecture was able to abstract higher-order patterns from these features as compared to simpler non-hierarchical models.

While these results already constitute an improvement compared to the current state-of-the-art analysis of RNA synthetic biology tools, we hypothesized that the use of pre-computed rational features as network input led to information loss that could inherently limit the predictive power of these models. Considering that possibility, we trained an MLP model solely on one-hot encoded sequence representations of our

toehold switches, eliminating potential bias introduced by a priori mechanistic modeling. We found that this sequence-based MLP delivered improved functional predictions based on $R^2$ and MAE metrics ($R^2$: ON = 0.70, OFF = 0.53, ON/OFF = 0.43) (Fig. 3d, e and Supplementary Fig. 9). These values represent a doubling of $R^2$ performance as compared to the MLP trained on rational features and a tenfold improvement in ON/OFF $R^2$ over the best individual rational feature used for previous linear models. When training for classification, our one-hot sequence MLP produced similarly improved AUROCs and AUPRCs of 0.87 and 0.36, respectively (Fig. 3f).

The improvement in performance when training on sequence-only inputs compared to rational features suggests that significant information loss occurs when performing thermodynamic calculations on toehold-switch sequences, a problem that may extend to other RNA synthetic biology tools in use today. The sequence-only MLP model dramatically outperformed a logistic regressor model trained on the same one-hot sequence input (Fig. 3d–f), further supporting the hypothesis that improved accuracy of our sequence-based MLP arises from learned

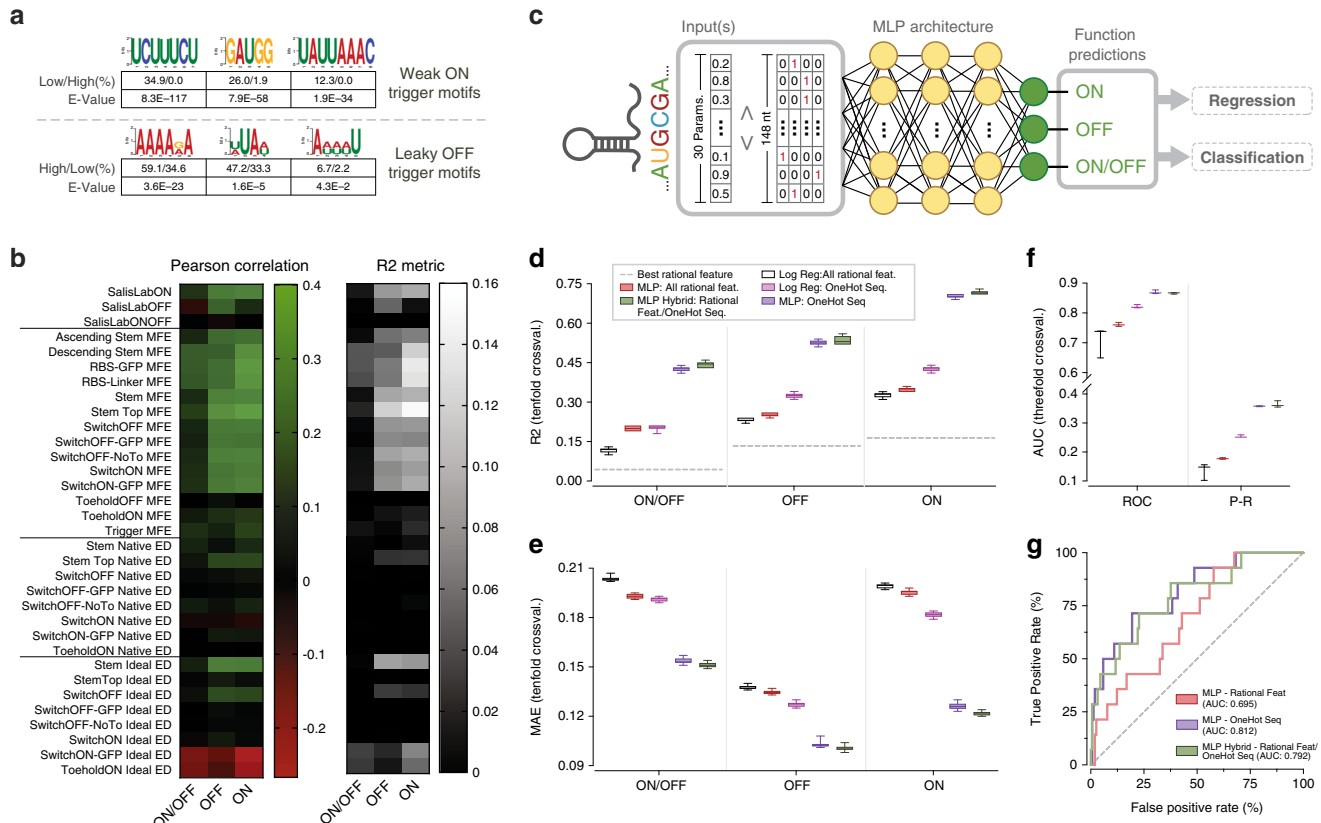

**Fig. 3 Analysis of toehold-switch performance using multilayer perceptron (MLP) models. a** Sequence logos for k-mer motifs discovered to be disproportionately represented in weakly induced switches (low ON) and leaky switches (high OFF), functional proportions, and *E*-values. **b** The Pearson correlation (left, |max| = 0.4) and $R^2$ metric (right, |max| = 0.16) for 30 state-of-the-art thermodynamic features and obtained RBS Calculator v2.1 outputs. **c** Base architecture of investigated MLP models, featuring three fully connected layers. For training in regression mode, three different outputs were predicted (ON, OFF, ON/OFF), whereas for classification training, only a single binary output based on ON/OFF (threshold at 0.7) was predicted. **d** Box-and-whisker plots for $R^2$ between experimental and regression-based predictions for best-performing rational features, logistic regression models and MLPs using tenfold cross-validation (test sets randomly selected from quality control process #2, QC2 in Supplementary Fig. S13 and Supplementary Table 1). **e** Box-and-whisker- plots for mean absolute error (MAE) between experimental and predicted values for these same models. **f** Box- and-whisker plots for the area under the curve (AUC) of the receiver–operator curve (ROC) and the precision-recall curve (P–R) in classification-mode predictions compared to experimental values using threefold cross-validation (test sets randomly selected from quality control process #2, QC2 in Supplementary Fig. S13 and Supplementary Table 1). In both regression and classification, the one-hot encoded sequence MLP delivered top-in-class performance without using pre-computed thermodynamic or kinetic metrics. **g** ROC curves of pre-trained MLP classification models validated with an unseen 168-sequence external dataset from Green et al.[2]. For all box-and-whisker plots, the horizontal line indicates the median, box edges are at the 25th and 75th percentiles, and whiskers indicate the smaller of either 1.5 × IQR or max/min. All source data are provided as a Source Data file.

hierarchical nonlinear features extracted directly from RNA sequences. Concatenating both the rational features and the one-hot representation into a combined input gave a small but significant improvement in regression mode ($\Delta R^2 \approx 0.025$ and $\Delta$MAE $\approx -0.0025$, $P < 0.05$ for all six comparisons, two-tailed *t* test), but no significant improvement for AUROC or AUPRC when in classification mode (Fig. 3d–f). These results suggest that while the use of rational features may facilitate the abstraction of potentially relevant information of toehold-switch function, the one-hot sequence-only MLP model can recover such information without a priori hypothesis-driven assumptions built into the model if given sufficient training data.

In order to evaluate the degree of biological generalization in our sequence-only MLP model, we performed two additional rounds of validation. First, we iteratively withheld each of the 23 tiled viral genomes in the dataset during training and predicted their function as test sets, resulting in a 0.82–0.98 AUROC range (average 0.87; Supplementary Fig. 10), similar to previous results from our sequence-only MLP. We then carried out an external validation on unseen data from a previously published dataset of

168 characterized toehold switches[2] that had been collected under different experimental conditions. Our MLP models achieved an AUROC of 0.70, 0.81, and 0.79, when trained on rational features, one-hot sequence, and concatenated inputs, respectively (Fig. 3g). The improved performance observed when training the models directly on nucleotide sequence rather than thermodynamic features, even for an external dataset, suggest a competent degree of biological generalization and supports the value of modeling RNA synthetic biology tools using deep-learning and high-throughput datasets, removing the current assumptions of mechanistic rational parameters.

**Predictive performance of higher-capacity models.** Having explored a baseline deep-learning architecture, we next sought to determine whether training our dataset on higher-capacity convolutional neural networks (CNN) and long short-term memory (LSTM) recurrent neural networks could increase our predictive ability. CNN and LSTM models have been applied to a variety of biological datasets in recent years, and have been cited as being

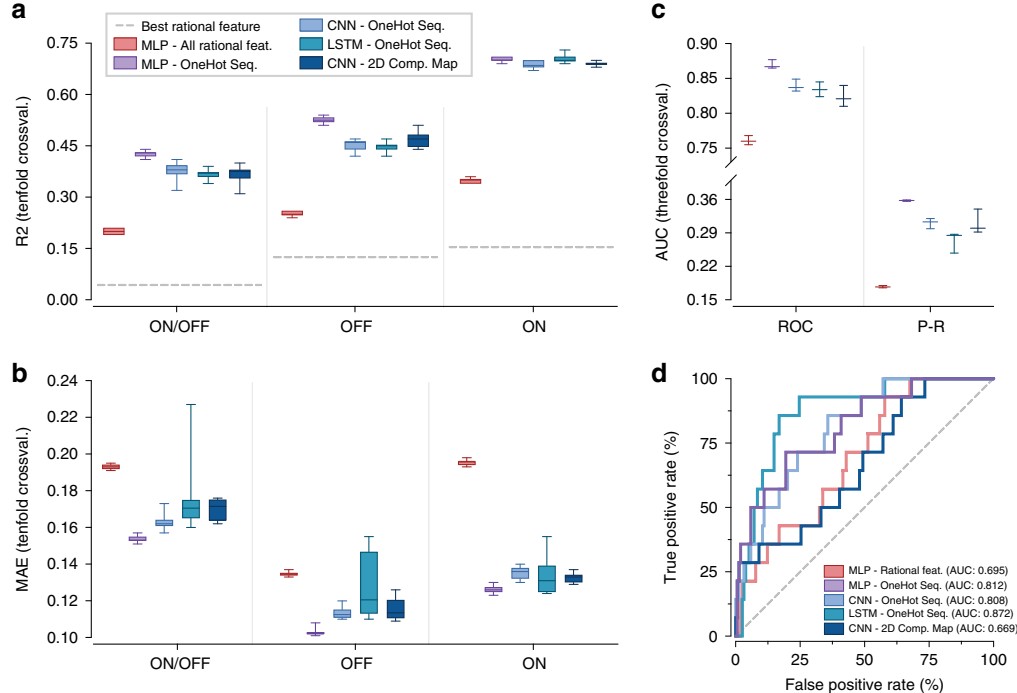

**Fig. 4 Evaluation of neural network architectures with increased capacity.** Performance metrics for convolutional neural networks (CNN) and long short-term memory (LSTM) networks trained on one-hot encoded toehold sequences, as well as a CNN trained on a two-dimensional, one-hot encoded sequence complementarity map. All models are compared to the previously reported MLPs trained on the 30 pre-calculated thermodynamic features and one-hot toehold sequences. For regression-based predictions, **a** shows box-and-whisker plots for $R^2$ metric, while **b** shows box-and-whisker plots for mean absolute error (MAE) for all models. In the case of classification-based predictions, **c** shows box-and-whisker plots of the area under the curve (AUC) of the receiver–operator curve (ROC) and the precision-recall curve (P–R) for all tested models. In both regression and classification, the one-hot encoded sequence MLP delivered a top-in-class performance as compared to higher-capacity deep-learning models. **d** ROC curves of pre-trained higher-capacity classification models validated with an unseen 168-sequence external dataset from Green et al.[2]. For all box-and-whisker plots, the horizontal line indicates the median, box edges are at the 25th and 75th percentiles, and whiskers indicate the smaller of either 1.5 × IQR or max/min. All source data are provided as a Source Data file.

particularly adept at recognizing motifs and long-range interactions in nucleotide sequence data[10,18–20,40–44]. We trained a CNN on a one-hot sequence input, an LSTM on a one-hot sequence input, and a CNN on a two-dimensional, one-hot complementarity map representation input (see "Methods" for complete descriptions of all models). Upon evaluating both the $R^2$ and MAE in regression mode and the AUROC and AUPRC in classification mode for these models (Fig. 4a–d), we concluded that these neural network architectures did not lead to superior predictive models, as compared to the sequence-based, three-layer MLP described previously. In these cases, increased model capacity led to under- or over-fitting, requiring additional training examples or improved fine-tuning to accelerate effective trainings.

**Visualizing learned RNA secondary-structure motifs.** One significant challenge of using deep learning to predict biological function is the inherent difficulty in understanding learned patterns in a way that helps researchers to elucidate biological mechanisms underlying model predictions. Recent work has been developed to visualize sequence features by mapping learned convolutional filters to biologically relevant sequence motifs[45,46]. Additional methods have been established to address how models link biological theory, including alternative network architectures[47], and the use of saliency maps[48,49], which reveal the regions of input that deep-learning models weigh most heavily and therefore pay the most attention to when making predictions. While saliency maps have been previously used to visualize model attention in one-hot representations of sequence data[10,18,20,48],

such implementations focus only on the primary sequence and have not been developed to identify salient secondary-structure interactions, which are especially relevant in the operation of RNA synthetic biology elements. Furthermore, prior work related to RNA secondary structure prediction using deep learning[50] has not utilized saliency techniques to highlight relevant secondary-structure regions that lead to improved function in RNA sensors. Instead, visualized representations have been constrained to predetermined structures based on the predictions of thermodynamic models[43,44], whose abstractions we have found cause significant information loss.

We sought to visualize important RNA secondary structures learned by our neural networks as it relates to biological function. To achieve this visualization, we trained a CNN on a two-dimensional nucleotide complementarity map representation (Fig. 5a) to allow for attention pattern visualization in this secondary-structure space. Each position in this complementarity map corresponds to the potential pair between two nucleotides, indicating its identity with a one-hot encoding (G–C, C–G, A–U, U–A, G–U, U–G, or a canonically unproductive pair). We hypothesized that by training deep networks on such a representation of RNA sequences, it would be possible to generate saliency maps revealing learned secondary structures as visually intuitive diagonal features. Importantly, because the complementarity map is unconstrained by a priori hypotheses of RNA folding (similar to our sequence-based MLP models), we anticipated this approach to be able to identify secondary structures that might be overlooked by commonly used thermodynamic and kinetic algorithms, such as NUPACK and Kinfold.

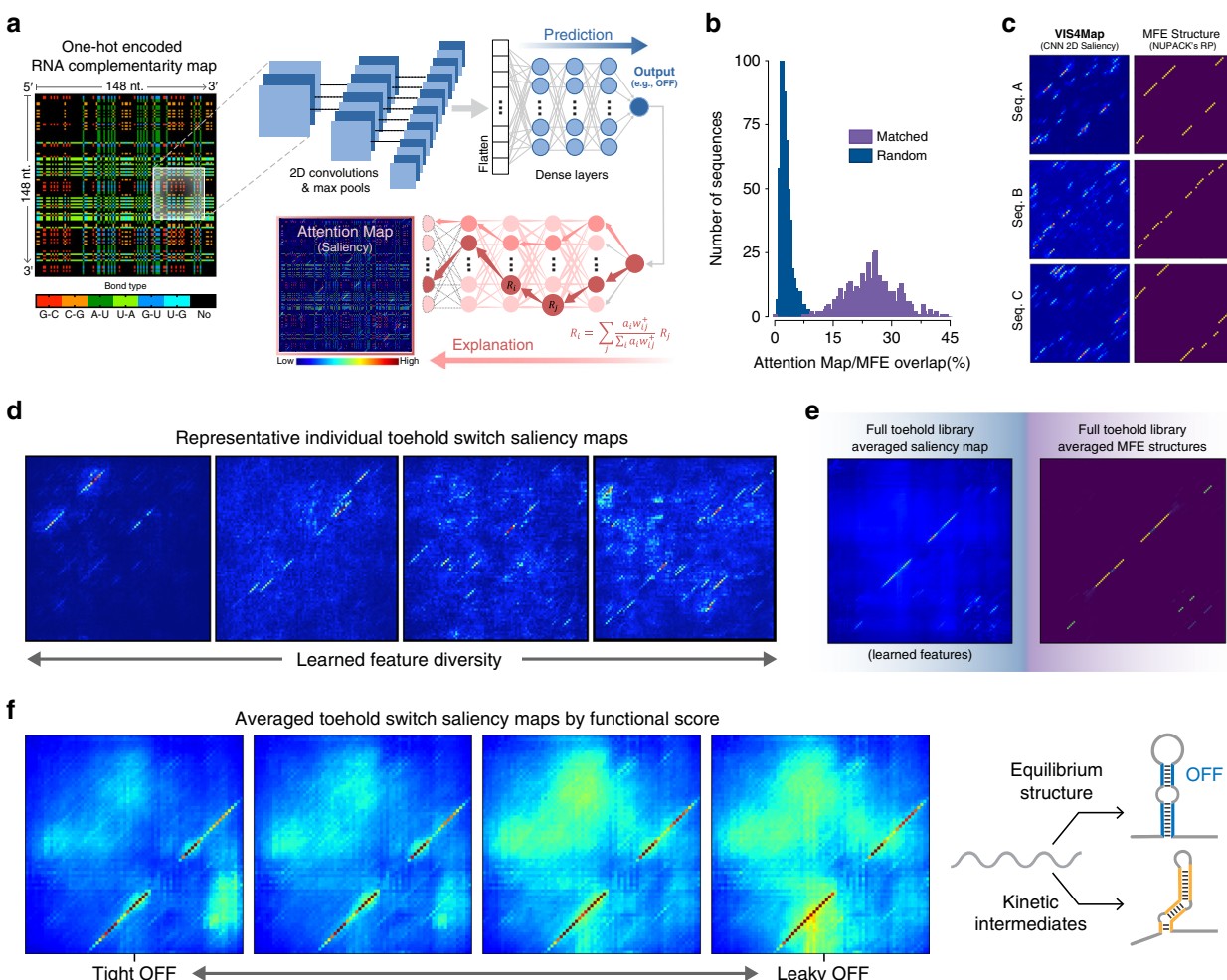

**Fig. 5 VIS4Map: visualizing learned secondary structures with complementarity matrices. a** A simplified schematic of the convolutional neural networks (CNN)-based architecture used to generate toehold functional predictions with network attention visualizations. The system receives a one-hot encoded, two-dimensional (2D) sequence complementarity map as input, followed by three 2D convolutional/max-pooling layers, a flattening step, and finally a set of dense layers. After output generation (e.g., OFF), a gradient-weighted activation mapping is performed to visualize activation maximization regions responsible for delivered predictions (VIS4Map). **b** Histograms of the percentage overlap between VIS4Maps generated from a CNN pre-trained to predict minimum free energy (MFE) using 120-nt RNA sequences and MFE maps generated by NUPACK. When analyzed using 500 random test-set sequences, the distributions of correctly matched and randomly assigned maps are distinct with increased percentage overlap from matched samples as compared to unmatched. **c** Examples of saliency VIS4Maps compared with their corresponding MFE structures as predicted by NUPACK for three randomly selected 60-nt RNA sequences. See Supplementary Fig. 11A for additional examples with 120-nt RNA sequences. **d** Four representative VIS4Map examples of randomly selected 118-nt RNA toehold-switch sequences from an OFF-predictive CNN model. **e** Averaged VIS4Maps of 10,125 randomly selected toehold-switch RNA sequences from our library test set processed with our OFF-predicting CNN model (left) and compared their corresponding averaged MFE maps obtained using NUPACK (right). **f** Averaged VIS4Maps of the 10% most accurately predicted switches sorted by quartile from lowest OFF (tight) to highest OFF (leaky); inset for the toehold and the hairpin stem. After contrast enhancement of averaged VIS4Maps to visualize sparsely distributed secondary structures, a noticeable increase in structures outside of the prominent equilibrium-designed switch hairpin structure appears to correlate with increased toehold leakiness. A toehold-switch schematic (right) is shown to denote how incorrectly folded and potentially weaker kinetically stable intermediate structures might compete with the correctly folded structure that is designed to be reached at equilibrium. All source data are provided as a Source Data file.

To first validate whether our visualization approach could capture any meaningful RNA structure features, we trained a CNN to predict NUPACK MFE values from a complementarity map representation of a randomly selected in silico RNA sequence dataset. Because NUPACK's calculated MFE is directly determined by a predicted RNA secondary structure, we anticipated that a CNN undergoing this training would likely pay attention to secondary-structure features, a situation that was confirmed through visualization of individual attention maps (Fig. 5b, c). Indeed, the saliency maps generated from a CNN trained on a complementarity map input contained primarily diagonal features that showed a statistically significant degree of agreement with the predicted MFE

structures from which NUPACK based its MFE calculations (Fig. 5b, c and Supplementary Fig. 11). Additionally, we found that the use of a complementarity map input improved the CNN's predictions of MFE from $R^2 = 0.6$ to $R^2 = 0.74$ compared with a one-hot sequence input (Supplementary Fig. 11). Hence, without prior knowledge of the algorithm or parameters NUPACK uses to calculate MFE, our CNN was able to learn similar abstractions as NUPACK, which we then used to intuitively visualize underlying relevant RNA secondary structures utilizing our complementarity map input representation. We named this approach for interpreting RNA deep-learning models Visualizing Secondary Structure Saliency Maps or VIS4Map.

Encouraged by our CNN's ability to elucidate putative RNA secondary-structure features directly from in silico-generated training data, we applied VIS4Map to our entire experimental toehold-switch function dataset. When trained on a complementarity map representation both in regression mode and classification mode (Fig. 5d), VIS4Map significantly outperformed an MLP trained on rational thermodynamic features. However, VIS4Map did not significantly outperform an MLP trained on a one-hot sequence input (as was the case when predicting NUPACK MFE). The failure of VIS4Map to improve predictions compared with a simpler three-layer MLP model likely results from over- or under-fitting of the higher-dimensional input, similar to the case of our other higher-capacity models (Fig. 4a–d). Encouragingly, nonetheless, we found that saliency maps produced by this CNN model displayed clear diagonal secondary-structure features (Fig. 5d). These structures appear to span from hybridization between the toehold and the ascending stem, to hybridization between the descending stem and the downstream linker. We confirmed the biological relevance of these features by averaging saliency maps and finding that the shared structures corresponded to the designed on-target structure of the switch hairpin (Fig. 5e). We further analyzed learned features outside of the designed equilibrium structure by sorting saliency maps using the toehold-switch OFF signal (Fig. 5f and Supplementary Fig. 12). We found that for leakier (high OFF) switches, the CNN identified a high degree of salient off-target secondary structures that could compete with the main hairpin stem and thereby expose the RBS, whereas for tight (low OFF) switches, the CNN identified fewer competing off-target secondary structures. In the context of general riboregulator behavior, these findings support the hypothesis that leaky expression from an RBS repressed by secondary structures can be caused by the misfolding of the repressive structure into less stable kinetic intermediate conformations[35,38] (Fig. 5f, right).

The fact that VIS4Map was able to identify both equilibrium and kinetically stable RNA secondary structures indicates a remarkable ability to uncover biologically relevant information, which in this case supports currently postulated hypotheses on prokaryotic translation initiation. Importantly, the identified secondary-structure features could not have been visualized using the one-hot sequence representation commonly associated with saliency maps[10,18,20]. These findings compound to the advantage of using sequence-only deep-learning approaches for analyzing RNA synthetic biology tools. Outside of toehold switches and other synthetic RNA systems, we anticipate VIS4Map will be broadly useful for the discovery of previously unknown equilibrium or kinetically stable structures contributing to RNA biology that are not predicted by current mechanistic RNA structure models.

## Discussion

Here, we presented a high-throughput DNA synthesis, sequencing, and deep-learning pipeline for the design and analysis of a programmable RNA switch. Having produced a toehold-switch dataset ~100-fold larger than previously published as a model system for investigating synthetic RNA response elements[2–6,15,16], we demonstrated the benefits of using deep-learning methods that directly analyze sequence rather than relying on calculations from mechanistic thermodynamic and kinetic models. This approach resulted in a tenfold improvement in functional prediction $R^2$ over an ensemble of commonly used thermodynamic and kinetic features. Moreover, the validation of our deep-learning models on an external previously characterized dataset, as well as the holdout prediction of every individual viral genome in our dataset, further

demonstrated the robust biological generalization of our models. Collaborative efforts by Valeri et al.[17] also extended our work, with the implementation of a natural language modeling approach and the de-novo design and testing of toehold switches using deep-learning models.

As with most work in RNA synthetic biology, all previous attempts to improve toehold-switch functionality have relied on the guidance of mechanistic thermodynamic modeling and low-throughput datasets[2–8,15,16]. Too frequently, rational design rules fail to give meaningful predictions of function for RNA-based synthetic systems. The results presented here suggest that the biological processes underlying RNA biology may be more complex than current state-of-the-art analyses take into account and that high-throughput DNA synthesis, sequencing, and deep-learning pipelines can be more effective for modeling said complexity. Combining improved predictions with enhanced understanding, our VIS4Map method further allowed us to visualize the equilibrium and kinetic secondary-structure features that our deep-learning models identified as important to the leakage of the switch OFF state. While secondary structures identified by NUPACK, Kinfold, and other rational mechanistic models are limited by predefined abstractions, which may cause significant information loss, our approach explored sequence space in an unrestricted manner and analyzed all possible RNA secondary structures. VIS4Map could prove useful for identifying complex secondary-structure information that might otherwise be ignored by simplified physical energetic models of RNA folding.

The dataset reported here also represents an extensive repository of characterized toehold switches, which could be used to accelerate the development of engineered riboregulators and future cell-free diagnostics[3,4,15,16]. These switches tile the entire genomes of 23 pathogenic viruses of high clinical importance, as well as hundreds of human transcripts, including many that are differentially expressed in cancerous phenotypes[51,52]. The total cost of our flow-seq pipeline equates to ~$0.08 per measurement, suggesting that the benefits of high-throughput design and assaying of RNA synthetic biology tools could be made widely accessible. We hope that this work will encourage the use of high-throughput data collection for the training of deep-learning systems, paired with more interpretable neural network architectures unrestricted by thermodynamic or kinetic secondary-structure models for improved prediction and insight generation in RNA synthetic biology.

## Methods

**Toehold-switch architecture selection**. The first-generation toehold-switch architecture from Green et al.[2] was selected in order to maximize the sequence variability in switch regions contributing to secondary structure. Where in later designs, the trigger RNA only unwound a fraction of the stem[2–4], in this earlier design, the entire hairpin stem was variably complementary to the trigger, increasing the diversity of characterized RNA hairpins (Fig. 1). An alternative fused ON state was also utilized. Normally, toehold switches detect the presence of a separate trigger RNA transcribed in trans to the OFF-state switch mRNA. However, for the testing of a large library of toehold-switch pairings, a two-plasmid system becomes intractable because each switch is designed around a specific cognate trigger. A two-plasmid system can also increase stochasticity caused by copy number variability. Green et al.[2] found a strong positive correlation between conditions when the trigger is fused to the switch and conditions when un-fused, separate triggers are transcribed in excess. We confirmed this correlation ourselves on a subset of 20 toehold switches by comparing the signal from the alternative fused ON state used in our library to the measured ON/OFF results from Green et al.[2]. Green et al. did not report separate ON and OFF measurements but stated that due to a low switch plasmid copy number, their OFF state rarely exceeded background autofluorescence, meaning that their reported ON/OFF ratios are approximations of ON-state measurements. The resulting comparison of signal from the alternative fused ON state we measured and the un-fused ON/OFF ratio measured by Green et al. using a two-plasmid system resulted in a Pearson $R = 0.8567$, as seen in Supplementary Fig. 1b. Thus, the ON state of the switch can be reliably approximated by fusing the trigger RNA to the 5' end of the switch mRNA

using a constant, unstructured linker sequence (Fig. 1 and Supplementary Fig. 1a), allowing for the direct synthesis of trigger-switch cognates on a single plasmid.

**Library trigger sequence selection.** Viral genomes were obtained on November 6, 2018, from https://www.ncbi.nlm.nih.gov/genome/viruses/. Each retrieved genome was tiled 30 bp at a time (the trigger length), with a stride of 5 bp, spanning the respective genome. Human transcription factors were obtained using ENSEMBL 94 BioMart[53] utilizing the Gene Ontology term GO:0044212 (transcription regulatory region DNA binding). The coding region of each transcription factor was tiled 30 bp at a time with a stride of 10. A remaining portion of the designs (~10,000) was based on random 30 bp triggers.

**Toehold library synthesis.** We designed 244,000 toehold-switch variants using 230-bp oligos, which were ordered and synthesized by Agilent. For each toehold-switch variant, the oligo was designed containing the following sequence components in order from 5′ to 3′: 20 nt of common backbone, a T7 promoter, the 30-nt trigger sequence, a 20-nt unstructured linker, the 12 nt toehold, the 18-nt ascending stem, a 11-nt SD-containing stem, the 18-nt descending stem including the start codon, a 21-nt AA linker, and the first 15 nt of the GFP gene. A schematic of the design can be found in Supplementary Fig. 1a. In the previous validation of the fused-trigger approach by Green et al.[2], only part of the trigger was fused to avoid recombination of long repeated sequences, but the nature of our flow-seq pipeline allowed us to avoid this issue since the integrity of all variants was confirmed after measuring fluorescence through NGS. The oligos were received at a stock amount of 10 pmol, which we diluted in 500 μL TE buffer for a working concentration of 20 nM. Of this working stock, 0.25 μL was used in 50 μL qPCR reactions using NEB Q5 polymerase 2×MM with 50 nM final concentration of appropriate primers. Two separate amplifications were done from the working stock of the oligo library for the ON and OFF states, respectively. One amplification, for the ON state, used a primer hybridizing to the 5′ common backbone region. The resulting insert contained both the switch RNA module and the trigger attached to its 5′ end. The second amplification, for the OFF state, used a primer hybridizing to the 20-nt unstructured linker and included a T7 promoter and the 5′ common backbone region in its tail. The OFF-state insert contained only the switch RNA module without the trigger module attached. See Supplementary Fig. 1a for a full schematic of the amplification scheme. A third amplification linearized a ColE1 plasmid backbone for subsequent ligation. This backbone was the same ColE1 backbone as was used in Green et al.[2] for transcribing trigger RNAs, but with a GFPmut3b-ASV gene inserted. All amplicons were cleaned from their reaction buffers by using carboxyl-coated magnetic beads[54] (protocol 4.3): 1× concentration of beads to clean the longer linear backbone product, and 2× bead concentration to clean the smaller insert products. Both inserts were ligated separately into the ColE1 backbone in front of the GFPmut3b-ASV gene using golden gate cloning, as follows. The linearized plasmid backbone was diluted to 500 ng of the total mass. The ON or OFF insert was added according to a 1:1 molar ratio of insert to the plasmid backbone. The inserts and backbone dilutions were prepared into 50 μL of ligation reaction volumes, containing 5 μL of NEB buffer 3.1, 5 μL of T4 ligase buffer, 1 μL of BsmBI, 0.5 μL of Dpn1, 1 μL of T4 ligase, and any remaining volume with nuclease-free water. The 50 μL reaction was placed into a thermocycler for 100 cycles of two steps: 16 °C for 10 min and 37 °C for 10 min. A final enzyme inactivation step at 65 °C for 15 min was done. The ligation products were precipitated out of their reaction buffers using ethanol precipitation. The 50 μL of ligation reactions were added to 1.5-mL Eppendorf tubes containing 150 μL of pure ethanol, 5 μL 0.3 M sodium acetate (pH 5.2), and 1 μL of GlycoBlue. Tubes were left on dry ice for 20 min and then immediately placed in a 4 °C tabletop centrifuge and spun at max speed for 30 min. Tubes were decanted, and 175 μL of 70% ethanol was added to the tube containing the pellet. Tubes were spun at max speed for 5 min. Tubes were then removed from the centrifuge, decanted, and allowed to dry for 15 min. Ligation products were then eluted in 4 μL of TE buffer. For initial library transformation, 50 μL of EclonI Supreme cells were given the full 4 μL of ligation product elution and electro-transformed. Transformation efficiencies exceeding 10[7] CFU/mL were achieved, and the expanded cells were harvested using a MaxiPrep kit (Qiagen). The resulting pool of plasmids was then electroporated into BL21 star *E. coli*, where transformation efficiencies exceeding 10[6] were achieved.

**Flow-seq pipeline.** Induction was achieved by expanding BL21 cells overnight at 37 °C in LB media with carbenicillin (carb) selection and then diluted 50× into fresh media. After the cells reached OD600 of 0.3 at 37 °C (~2 h of growth), 0.2 mM IPTG was added, and the cells were allowed to express for another 3 h at 37 °C. The cells were then moved to room temperature and sorted on a Sony SH800 FACS machine (with accompanying Sony Sorter Software SH800S) using four bins (i.e., gates), with each bin spanning approximately one order of magnitude of GFP fluorescence (Supplementary Fig. 2a). To facilitate comparison between the ON and OFF libraries and to ensure both would be measured on the same scale of GFP signal, we utilized two control plasmids to anchor the lowest and highest GFP expression levels for sorting. A high-performing switch from Green et al.[2], referred to by the authors as Switch #4, was cloned both in its OFF state and in the modified, fused-trigger ON state. The Switch #4 ON state expressed at very high

levels in our assay, and when compared to the full library distributions of all ON and OFF variants, this control switch marked the highest total levels of GFP signal (Supplementary Fig. S2). We thus used the Switch #4 distribution to demarcate the highest bin of activity. We used the pUC19 plasmid as a negative control to mark the lowest bin of GFP signal (Supplementary Fig. 2a) since it does not contain GFP. The number of bins used was chosen based on a preliminary study of our flow-seq pipeline characterizing a panel of 20 previously published switches from Green et al.[2] (Supplementary Fig. 1d). Clonal toehold variants showed a normal distribution of intensity that roughly spanned an order of magnitude (as seen for Switch #4 in Supplementary Fig. 2), and no difference in measured flow-seq signal was observed when sorting on four or eight bins, suggesting that four bins were sufficient to accurately measure fluorescence across four orders of magnitude in high-throughput (Supplementary Fig. 1e). Nonetheless, it should be noted that at lower read-sampling thresholds, artifacts were observable at the borders of the four bins (Supplementary Fig. 13, see "Library Quality Control" for a further discussion of these artifacts).

We found the GFP expression levels for each library for ON and OFF variants contained a full spectrum of activity between the levels of the control plasmids utilized (Supplementary Fig. 2). For example, faulty OFF-state switches with high degrees of leaky GFP expression yielded fully ON-like states with maximum GFP intensity, likely because incorrect folding resulted in leaving the RBS exposed. Similarly, faulty ON-state switches had the lowest, negative control levels of GFP intensity, presumably because those variants' triggers could not efficiently unfold the switch hairpin stem, thus leaving the RBS sequestered.

Approximately 10 million events were sorted for each gate and for each library. Cells in collected bins were diluted 10× into fresh LB media with carb selection and allowed to expand overnight at 30 °C. The expanded cells were then harvested using a MaxiPrep kit (Qiagen).

**Deep sequencing and read count analysis.** Plasmid collected from sorted cells was amplified using NEB Q5 polymerase 2×MM and primers targeting the common backbone region upstream and downstream of the variable toehold region. The resulting 184 bp (OFF) or 224 bp (ON) PCR products were then analyzed by NGS using a MiSeq or NextSeq instrument (Illumina). Raw paired-end sequencing reads were quality filtered and merged with PEAR 0.9.1. The distribution of GFP signal in the flow-sorting data displayed in Supplementary Fig. 2a is highly imbalanced for both the ON and OFF libraries. A large fraction of the oligo library pool contained incorrectly synthesized oligomers. These were largely truncated products lacking a start codon, lacking an SD sequence, or containing a frameshift that we would expect to lead to low GFP signal. We estimate that at least 50% of the cells that we sorted contained such a truncated variant, and most of these ended up in the lowest bin. Thus, only sequences matching our intended designs were retained for further analysis. For the ON and OFF libraries, respectively, 10,390,207 reads and 20,788,966 reads were mapped to a correct switch sequence. The final ON and OFF datasets seen in Fig. 2 are notably less skewed than the flow-sorting data seen in Supplementary Fig. 2, thanks to the exclusion of reads corresponding to incorrectly synthesized switches.

Frequencies of each variant were tabulated for each cell-sorted bin and normalized to the total reads per bin. Each variant's functional value was computed as the weighted mean of its normalized frequencies across all bins. Because each library was sorted using the same gates established by the control plasmids (see "Methods" for "Flow-seq pipeline"), and since each library spanned a remarkably similar range of minimum and maximum GFP intensity (Supplementary Fig. 2a, b), we scaled the ON and OFF values for each variant to fall between [0, 1]. A value of 0 was given to a variant if all corresponding reads were found only in the lowest bin and a value of 1 if all corresponding reads were found only in the highest bin. An ON/OFF metric was calculated by subtracting these individuals ON and OFF signal metrics (Fig. 1), which resulted in values between [−1, 1].

**Library quality control.** A second biological replicate of our flow-seq pipeline was carried out that produced 60,800 ON measurements, 98,295 OFF measurements, and 30,101 ON/OFF ratio measurements where both ON and OFF were available for the same switch. The $R^2$ and MAE between our two datasets were calculated at different read count thresholds. Based on the results (Supplementary Fig. 3), five different QC thresholds were established, some of which also included standard deviation cutoffs (Supplementary Table 1 and Supplementary Fig. 13). QC1 and QC2 contained OFF data with significantly worse $R^2$ compared to QC3, QC4, and QC5, but only QC1 contained OFF data with worse MAE. We determined that the inter-replicate drop in $R^2$ for OFF values was mainly due to the skewness of the data—indeed, the OFF data consistently showed worse $R^2$ values than the ON data throughout the paper, despite having consistently better MAE values. Therefore, we chose to trust in the inter-replicate MAE values more than the inter-replicate $R^2$ metric for the OFF data.

To further evaluate the different QC levels, the most stringent data (QC5) were withheld as a test set, and an MLP fed a one-hot representation of the toehold sequence was trained on the four lower-QC levels. The results for predictive $R^2$ showed QC1 to be of significantly inferior quality, but QC2, QC3, and QC4 to be of roughly similar quality (Supplementary Fig. 4). This result was consistent with the fact that inter-replicate MAE and $R^2$ were notably worse at the QC1 count threshold compared with the read count thresholds contained by QC2, QC3, and

QC4 (Supplementary Fig. 3). The QC2 dataset gave the best predictive results by a small margin and was also significantly larger than QC3 or QC4 (Supplementary Table 1). With these analyses in mind, QC2 was chosen as the final threshold for inclusion in our dataset. Within the measured ON/OFF ratios in the QC2 dataset, 40,824 had triggers of viral origin, 47,005 had triggers of human origin, and 3705 had randomly generated trigger sequences.

Artifacts of the flow-seq pipeline are also clearly visible in lower-QC datasets (Supplementary Fig. 13). These manifest as "spikes" of intensity at the borders of the sorting gates, corresponding to an overrepresentation of variants with reads in only one bin. As read count thresholds increase and sampling improves, such variants become rarer—the artifacts are visible in QC1 and QC2, less visible in QC3 and QC4, and largely absent in QC5. Given the possibility that models trained on different data inputs (sequence-only vs biophysical) might fit data with such local distortions to differing degrees, we also analyzed the performance of existing biophysical models and neural network models trained on biophysical parameters against all datasets at QC levels above QC2, with QC5 being the most stringent set that did not contain any apparent sorting artifacts (Supplementary Fig. 4). We did not observe a meaningful improvement in $R^2$ accuracy using an MLP trained on the biophysical rational parameters at QC levels 1–4 and then tested on QC5. Neither were significant improvements in $R^2$ correlation seen between the data and individual biophysical parameters at higher QC levels (Supplementary Fig. 16). We are therefore confident that at the quality control level chosen for the final dataset (QC2), the sorting artifacts did not differentially impact model performance.

**Cell-free switch validation**. Eight of the best-performing switches (ON/OFF > 0.97) and eight of the worst-performing switches (ON/OFF < 0.05) were synthesized as PCR products, as previously described[3,4]. Briefly, they were ordered as single ultramer oligos (IDT) without the trigger fused, from the T7 promoter to the first 36 nt of the common linker and GFP sequences. These were added to a GFP gene by a single PCR amplification step. Triggers were in vitro transcribed from separate oligos that contained the antisense sequence and the antisense T7 promoter, to which the sense strand of the T7 promoter was annealed. Trigger RNA was purified using an RNA Clean & Concentrator kit (Zymo), while switch DNA was purified using a MinElute kit (Qiagen). To a 5-μL PURExpress reaction were added 2U/μL Murine RNASe Inh, 5 nM of toehold-switch PCR product, and either no-trigger RNA or 10uM of trigger RNA. Measurements of GFP velocity can be found in Supplementary Fig. 5. The switches tested and their library assay measurements can be found in Supplementary Table 2.

**Calculations made with ViennaRNA, Kinfold, and the RBS calculator**. All thermodynamic MFE and ensemble defect calculations, as well as kinetic Kinfold calculations, were obtained using a custom-made python code including libraries from packages such as Biopython (ref.: https://github.com/biopython/biopython), ViennaRNA (ref.: https://github.com/ViennaRNA/ViennaRNA), RNAsketch (ref.: https://github.com/ViennaRNA/RNAsketch) and Pysster (ref.: https://github.com/budach/pysster). Calculations of thermodynamic rational parameters to include in our database were obtained from each basal 145-nucleotide toehold sequence by taking each basal 145-nucleotide toehold sequence and then isolating different sections (e.g., GGG, Trigger, Loop1, Switch, Loop2, Stem1, AUG, Stem2, Linker, Post-linker) into distinct subsequences with biological relevance for functional analysis (see Supplementary Fig. 1 and Supplementary Table 4). Minimum free energy (MFE) was calculated for all these sections using the previously ported python-based ViennaRNA Library. MFE calculation using ViennaRNA also specifies a secondary structure in dot-parens-plus notation (unpaired base = dot, base-pair = matching parentheses, and nick between strands = plus). Ideal structures are assumed to be connected and free of pseudoknots. These ideal secondary structures for such sections are:

SwitchOFF = '.............................((((((((…((((((……...))))))…))))))))'

SwitchOFF_GFP = '...............................((((((((…((((((……...))))))…))))))))..((…….(((((……))))).)))….'

SwitchOFF_NoTo = '((((((((…((((((……...))))))…))))))))..(((…….(((((……))))).)))….'

SwitchON = '…(((((((((((((((((((((((((…………….))))))))))))))))))))))))))…………………..'

SwitchON_GFP = '…(((((((((((((((((((((((((…………….)))))))))))))))))))))))))…………………..(((…….(((((……))))).)))….'

ToeholdON = '………………..(((((((((……………..)))))))))))'

Stem = '((((((((…((((((……...))))))…))))))))'

StemTop = '((((((……...))))))'

Ensemble defect as a rational parameter was calculated via ViennaRNA/NUPACK for each of the toehold switches in the above subsets of sequence regions: SwitchOFF, SwitchOFF_GFP, Switch_OFF_NoTo, SwitchON, SwitchON_GFP, ToeholdON, Stem, StemTop. This calculation used both the native (calculated from MFE) and the ideal (predefined above) dot–bracket representation for each sequence to assess the average number of nucleotides that are incorrectly paired at equilibrium. Thirty rational parameters were calculated for each toehold using these methods (fourteen MFE values, eight ideal ensemble defect values, and eight native ensemble defect values).

Kinetic analyses using Kinfold were run from the ViennaRNA package. The OFF-switch sequence was selected, spanning nucleotides 50 to 134 in Supplementary Table 4 from the start of the toehold to the end of the linker. Due to the large size of the toehold-switch RBS, Kinfold trajectories ran for 100–1000× longer than for RBS's previously analyzed relating to the RBS calculator in Borujeni et al.[35] (Supplementary Fig. 7b). Hence, our analysis was scaled down to the QC4 dataset (containing 19,983 total switches), with 100 Kinfold trajectories run for each switch with a maximum stopping time of $10^3$ arbitrary Kinfold units (au). The energy and time at each step of each trajectory were recorded. If the MFE structure was reached within $10^3$ au, it was assumed that the RNA would remain in the MFE structure for the rest of the $10^3$ au timeframe. From each energy trajectory spanning $10^3$ au, the average energy (in kcal/mol) was calculated by integrating the energy-time curve and dividing by $10^3$. For each switch, the following features were extracted: the mean and standard deviation of the average energy of its 100 sampled trajectories (Supplementary Fig. 7c), the ratio of the mean average energy to the MFE (Supplementary Fig. 7e), and the fraction of trajectories that reached the MFE structure within the analyzed $10^3$ timeframes (Supplementary Fig. 7d).

For predictions by the RBS calculator, an API was used to access the most recent publicly available version (2.1). Due to limiting computational costs, the QC3 dataset was used instead of the QC2 dataset. For each switch, the translation initiation rate (TIR) of the on-target start codon was predicted for both the ON and OFF states ("SwitchON_GFP" and "SwitchOFF_GFP" respectively in Supplementary Table 4).

**K-mer motif search**. In order to compare sequence-level motifs between the best and worst variants measured in our dataset, we performed a k-mer search for overrepresented sequence motifs at the tails of our observed functional values. We first filtered the variants for high quality, retaining those with a QC4 score or above. We then took the top and bottom 1000 variants based on the ON and OFF functional values, respectively. We utilized DREME[55] to test for enrichment or depletion of all possible subsequences of length 3–16 bases, using the indicated foreground and background frequencies. Results above the default E-value cutoff are shown in Fig. 3a and Supplementary Table 3.

**Deep-learning model architectures**. *MLP—rational features*: The multilayer perceptron (MLP) model based on rational features included a 30-feature input followed by three dense fully connected layers of 25, 10, and 7 neurons, respectively, with a rectified linear unit (ReLU) activation, batch normalization, and 30% dropout. This network was then fed to a final three-neuron layer (ON, OFF, ON/OFF) with linear activation for regression output, or to a final two-neuron layer (ON/OFF: binarized at +/− 0.7) with softmax activation for classification output.

*MLP—OneHot seq*: The MLP model based on the one-hot encoded full 145-nucleotide sequence input was achieved by using a flatten layer followed by three dense layers with ReLU activation, batch normalization, and 30% dropout. Dense layers used 128, 64, and 32 neurons, respectively. This network was then fed to a final three-neuron layer (ON, OFF, ON/OFF) with linear activation for regression output, or to a final two-neuron layer (ON/OFF: binarized at +/− 0.7) with softmax activation for classification output.

*MLP—hybrid rational features/onehot seq*: The ensemble MLP model was based on the rational features, as well as a one-hot encoded full 145-nucleotide sequence as input. To construct this model, two networks were assembled in parallel. The first network uses the same architecture for the MLP model with rational features, while the second network used the architecture of the MLP model for one-hot encoded 145-nucleotide sequences. Both networks were then concatenated and connected to a four-neuron dense fully connected layers with ReLU activation. This network was then fed to a final three-neuron layer (ON, OFF, ON/OFF) with linear activation for regression output, or to a final two-neuron layer (ON/OFF: binarized at +/− 0.7) with softmax activation for classification output.

*CNN—OneHot seq*: The convolutional neural network (CNN) model based on the one-hot encoded full 145-nucleotide sequence as input was achieved by direct feeding of the input to three convolutional layers with ReLU activation, batch normalization, and 30% dropout. The convolutional layers used had 32, 64, and 128 filters of size 3, respectively. Same-padding was used with L1 and L2 kernel regularization. The output from the convolutional layers was flattened and fed to two fully connected sequential dense layers of 16 neurons, each with ReLU activation, batch normalization, and 30% dropout. This network was then fed to a final three-neuron layer (ON, OFF, ON/OFF) with linear activation for regression output, or to a final two-neuron layer (ON/OFF: binarized at +/− 0.7) with softmax activation for classification output.

*CNN—2D complementarity map*: The CNN model based on the one-hot encoded categorical 2D complementarity-directional matrix from the full 145-nucleotide sequence as input was achieved by direct feeding of the input to three convolutional layers with ReLU activation, batch normalization, and 30% dropout. The convolutional layers used had 32, 64, and 128 filters of size 5 × 5, respectively. Same-padding was used with L1 and L2 kernel regularization. The output from the convolutional layers was flattened and fed to two fully connected sequential dense layers of 16 neurons, each with ReLU activation, batch normalization, and 30% dropout. This network was then fed to a final three-neuron layer (ON, OFF, ON/OFF) with linear activation for regression output, or to a final two-neuron

layer (ON/OFF: binarized at $+/-$ 0.7) with softmax activation for classification output.

*LSTM—OneHot seq*: The LSTM recurrent neural network model on the one-hot encoded full 145-nucleotide sequence as input was achieved by direct feeding of the input to a network with 128 recurrent units. The output of this was then connected to 100-neuron fully connected dense layer with ReLU activation, followed by batch normalization and 30% dropout. This network was then fed to a final three-neuron layer (ON, OFF, ON/OFF) with linear activation for regression output, or to a final two-neuron layer (ON/OFF: binarized at $+/-$ 0.7) with softmax activation for classification output.

All models were trained using a maximum of 300 epochs, considering a 20-epoch early stopping patience, which gets triggered upon lack of model improvement on the validation set. The batch size for all models was 64*(1 +ngpus), where ngpus is defined as the number of used graphic processing units during model training. All trained regression models were verified for reported metrics using tenfold cross-validation, while classification-trained models were evaluated on three shuffled test sets as indicated.

**Data balancing**. As part of a wide-reaching parameter search performed while optimizing our deep-learning models, we attempted four approaches to address the limitation of skewed OFF-state data (enumerated below). Interestingly, we found that these only gave at most very small improvements in model accuracy as measured by $R^2$, AUROC, or AUPRC (Supplementary Figs. 14 and 15). This suggested to us that by using un-transformed and unbalanced data, our models were already achieving nearly the best performance possible with those architectures. A trade-off of using unbalanced data is predictions often center around the total mean of the distribution. We utilized a variety of performance metrics, especially the AUPRC, to aid the interpretation of modeling unbalanced data. To compare the performance of various balancing strategies, we performed the following:

1. During regression, we transformed ON, OFF, and ONOFF data into a uniform distribution using sklearn QuantileTransform before training the model, and then transformed predicted test-set data back to their original values to calculate accuracy metrics. This transformation retained the rank-order of the data.
2. During regression, we balanced ON, OFF, and ONOFF data into a uniform distribution by splitting the data into twenty bins and randomly re-sampling data points from under-represented bins, done only for training and validation data. For withheld testing data, data points were randomly removed from overrepresented bins until a uniform distribution was achieved in order to show predictive performance across the range of data points.
3. During binary classification of ON/OFF, we balanced the high and low classes by randomly removing entries from the overrepresented lower class until the two classes contained the same number of entries.
4. During binary classification of ON/OFF, we balanced the high and low classes by randomly duplicating entries from the under-represented higher class until the two classes contained the same number of entries.

One factor that affected model accuracy was the cutoff for binary classification of ON/OFF. Increasing the cutoff for the high and low classes changed how imbalanced the ON/OFF data was, and had a significant effect on both AUROC and AUPRC. We carefully analyzed the implications of this technical consideration and described the decision we made to place the cutoff at ON/OFF = 0.7 (classifying the top 8.3% of ON/OFF values) in Supplementary Fig. 8.

**Complementarity matrix and VIS4Map**. Complementary maps were defined as a One-Hot Encoded Categorical 2D Complementarity-directional Matrix (total number of tensor dimensions = 3) constructed by defining columns and rows of the matrix as the position of potential complementarity between any two given pairs of nucleotides in a single RNA sequence. The value in each position is defined as a one-hot encoded categorical variable according to the Watson–Crick pairing of the two nucleotides defining that position. Nucleotide pairings are assigned the following category: G–C (6) = [0 0 0 0 0 0 1], C–G (5) = [0 0 0 0 0 1 0], A–U (4) = [0 0 0 0 1 0 0], U–A (3) = [0 0 0 1 0 0 0], G–U (2) = [0 0 1 0 0 0 0], U–G (1) = [0 1 0 0 0 0 0], NonWCpairs (0) = [1 0 0 0 0 0 0]. VIS4Maps were generated using a modified algorithm, attention, activation maximization and saliency map visualization for Keras (Keras–Vis, ref.: https://github.com/raghakot/keras-vis) with TensorFlow backend.

In this case, gradients were calculated from a regression model for all regions of the image to visualize what spatial features cause the predicted output to increase. To visualize the toehold regions that are mostly responsible for each prediction, small positive or negative gradients are highlighted using a normalization strategy. Given this information, such techniques allow us to generate heatmap-encoded saliency map images that spatially relate to the toehold regions in the complementarity map that lead to accurate predictions.

**Statistics and reproducibility**. All empirical experiments, including flow-seq assays used to produce our primary toehold-switch dataset, cell-free expression of candidate toehold switches, and FACS data collected for clonal populations of individual toehold switches and other constructs, were repeated at least once in order to verify the independent reproducibility of reported results. An exception is the smaller-scale flow-seq assay used to pilot our toehold-switch pipeline (see Supplementary Fig. 1d, e), which was not repeated.

All computational results, including reported cross-validation results as well as unreported architecture scans of our deep-learning models (including logistic regression models, MLP models, CNN models, and LSTM models), were repeated at least once in order to verify that the outputs could be independently reproduced.

**Reporting summary**. Further information on research design is available in the Nature Research Reporting Summary linked to this article.

## Data availability

A csv file containing the complete toehold-switch dataset is available from the same GitHub page as the code used in this work, and includes read counts for each sorting gate, derived flow-seq data, assigned QC scores, switch subsequences, and calculated rational parameter values. The same dataset as well as raw NGS seq read data can be obtained using GEO accession GSE149225. Any other relevant data can be obtained from the authors upon reasonable request. Source data are provided with this paper.

## Code availability

All custom code used in this work, including that used to train and test deep-learning models, perform saliency visualizations, and perform ViennaRNA/Nupack/Kinfold calculations, can be obtained from the following publicly accessible GitHub page: https://github.com/lrsoenksen/CL_RNA_SynthBio.

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

## Acknowledgements

This work was supported by the Defense Threat Reduction Agency grant HDTRA1-14-1-0006, the Paul G. Allen Frontiers Group, DARPA Synergistic Discovery and Design (BAA HR001117S0003) program, and the Wyss Institute for Biologically Inspired Engineering, Harvard University (N.A.M., A.S.G., L.R.S., and J.J.C.). N.A.M. was also supported by an MIT-TATA Center fellowship 2748460, while L.R.S. was also supported by CONACyT grant 342369/408970, and A.S.G. was also supported by DOE grant DE-FG02-02ER63445, NHGRI grant 5T32HG002295-12, and BIRT Fellowship T15LM007092. We thank Diogo Camacho for his aid in statistical analysis and data preparation methodologies. We also thank Max A. English, Nina M. Donghia, and Peter Q. Nguyen for their support in experimental activities of this work; Timothy Kassis for the helpful discussions and advice relating to the implementation of deep-learning model architectures; Max Schubert and Pierce Ogden for advice in optimizing library cloning; and Howard Salis for giving us access to his RBS calculator 2.1 API.

## Author contributions

N.A.M. and A.S.G. designed, constructed, and tested the toehold-switch database; N.A.M., A.S.G., and L.R.S. planned and performed experiments, wrote code, analyzed the data, and wrote the paper; J.J.C. and G.C. directed overall research and edited the paper.

## Competing interests

The authors declare no competing interests.
