## [Peer Review File · Nature Communications]

Reviewers' Comments:

Reviewer #1:

Remarks to the Author:

The paper presents a deep learning approach to resolve structure-function relationships in the context of toehold switches. In order to train a model that maps a toehold sequence to its ON/OFF foldchange the group generated around 100 000 toehold sequences of length 148. From the 148 bases roughly half of them remain fixed in order to deviate not too far from the canonical toehold structure.

First of all, generating and characterizing such a library is in itself a huge endeavor. If released to the public, the dataset would be very valuable for the scientific community for follow-up studies etc.

The trained neural network where the sequence is directly entered as a one-hot encoding shows a dramatic improvement in the predictive power over a network that only receives thermodynamic and kinetic features of the sequence or over purely biophysical models.

Apart from that demonstration, the work makes some effort to trace back the biophysical basis of why certain toehold sequences score high. In particular, the second input encoding in terms of the complementarity map is well motivated and allows us to learn something about the secondary structure features by computing saliency maps.

I have some remarks and questions that the authors should address:

1.) I have some concerns regarding the data normalization as given in the M&M section. It appears as if authors are normalizing the ON and OFF flow-seq data separately to their maximum (see Fig. 2 A & B). This would remove the actual foldchange and would only allow to rank structures relative to each other somehow. It would be not of a concern if both flow datasets are normalized to one common number. From flow-data in Fig S2 one finds (surprisingly) that the maximum of both, the ON and OFF library are roughly the same, which would indicate that the apparently performed separate normalization would not generate too much of a problem. However, simple calculus then says that the computed fold change is simply the log ratio of the raw data in the linear domain scaled by some constant (which might be easier to communicate). I do not believe that separate normalization is a good thing to do and the recurring theme in the paper that the biophysical models do not perform well and do better on Spearman correlations, might relate to this normalization issue. Authors should make that exposition of data processing more precise. Perhaps it is worthwhile to plot the raw flow-seq data where the not normalized ON and OFF distributions are shown on one scale.

2.) From Fig S2 it is also clear that the dataset is very unbalanced. For example, taking the ON dataset, the regression error will be dominated by structures that actually have a low ON level; same is true (and more severe) for the classification task. Hence, the network is provided with very few data points to learn switch features corresponding to high ON levels. That is consistent with Fig. S9, where one sees that the network (even the one-hot) poorly predicts high fold-change switches (more or less none is exceeding a ON/OFF ratio of 0.6 while experimental data goes up to one). The authors should state more clearly how they approached the problem that the dataset was unbalanced and be more explicit in the mentioned limitations.

3.) Staying with S9, one sees clear artifacts of the performed Flow-seq pipeline, where the pronounced grid lines are apparently due to those sequences that were exclusively detected in one FACS bin. Those accumulation points are obviously not physics but artifacts (such artifacts only start to disappear at QC5 according to Fig S13). Although authors discuss different QCs, I would like to see a more thorough and transparent discussion about those things and how they may affect the overall conclusions – in particular those that the biophysical models are performing so

poorly (in contrast to deep nets, biophysics cannot fit artifacts).

4.) Complexity and data-requirements for training strongly depend on the input dimension of the problem. Hence, in the traditional one-hot encoding I do not see any point in inputting also sequence positions that remain unaltered through the entire dataset. A network of reduced input dimension would train faster and would generalize better and would require less training data. Even for the complementarity map input one can envision such a reduced input representation. The author may do the corresponding experiment for the supplement.

5.) As ON and OFF level are also normalized and hence ON/OFF level follow a simple deterministic mapping, I do not see the point of trying to regress ON, OFF and ON/OFF values with the network. ON/OFF values can just be computed from the two ON and OFF network outputs.

6.) The references appear incomplete, i.e. there has been earlier work on optimizing riboswitches using machine learning (Groher A-C, et al. ACS Synthetic Biology, 2018). As for the presented comparison of cell-free versus in-vivo for riboregulators, there has been work supporting the made point that the rank order of switches is preserved but generally not their fold-change (e.g. Lehr F-X et al. ACS Synthetic Biology, 2019).

7.) It remains unclear to me, why authors did not take the next logical step of using the deep network as an inverse model to find novel toeholds with a high fold-change. This can either be done by sampling in combination with active learning / Bayesian optimization as has been done for protein engineering already (e.g. review of Yang et al. Nature Methods, 2019) or with some other back-tracking algorithms.

Reviewer #2:

Remarks to the Author:

This is an interesting paper on modeling RNA switches that can be published if a few minor issues are addressed:

- Why were only 4 bins used for the flow-seq experiment? And how quantitative are off-state measurements given background fluorescence? More detail on the sequencing strategy should be included in the main paper.
- The data in Fig. 2D suggest that flow-seq based measurements are not very correlated with in vitro fluorescence measurements. The fluorescence values vary over a 5-fold range even though all "good" switches presumably have similar fold change as measured by sequencing (ON/OFF > 0.97). Would it be better to compare flow-seq data to measurements performed with individual switches in vivo (i.e. flow cytometry using just a single switch at a time) and is the relative lack of quantitative agreement due to different switch behavior in cells and in vitro or due to facs-seq noise?
- What test set was used for the data shown in fig.3? please add this information. If it was randomly selected, it would be good to also try an alternative test set consisting of the most deeply sequenced switches. Such a test set (which should still be randomly distributed in sequence space) should help reduce noise due to the measurement. Even the best model presented only has $R^2=0.43$ for On/Off predictions and I wonder whether this is an artifact of noisy measurements.
- This statement is outdated: "One significant drawback of using deep learning to predict biological function is the inherent difficulty in understanding learned patterns in a way that helps researchers to elucidate biological mechanisms underlying model predictions." There is extensive recent work on visualizing DNA and RNA sequence features, mapping filter motifs to biologically and even using such models to guide sequence design for synthetic biology.

- The work shown in Fig. 5 is very interesting. The authors should use the model to explicitly predict on/off ratio for their switch test set and show that inclusion of secondary structure gives better results than sequence only models.
- I'm a bit confused by the following statements: "We sought to visualize RNA secondary structures learned by our neural networks in a manner unconstrained by thermodynamic modeling." Which is followed one paragraph later by "we first pre-trained a CNN to predict NUPACK MFE values from complementarity map representations of a randomly selected in silico RNA sequence dataset." Aren't these statements contradictory?
- The title is too general. Toehold switches are interesting but constitute only a very specific subset of RNA synthetic biology.

Response to Reviewers:

We greatly appreciate the time the referees invested in reviewing our paper. We have conducted additional experiments and revised the paper to address each of the points raised by the reviewers. We think these additions and revisions have strengthened the paper considerably. Of note, we investigated 42 new instantiations of our deep learning models in order to thoroughly characterize the effect of various data skew-compensation methods, sequence input parameters, and library quality control thresholds on performance. These data support our previous conclusions and show a robust analysis of our high-throughput dataset. We also performed additional experiments analyzing differences between toehold switch performance *in vivo* and *in vitro*, clarifying previously confusing discrepancies. Lastly, in order to thoroughly validate our experimental approach, we performed additional experiments applying our flow-seq pipeline to a new panel of previously characterized toehold switches. These results demonstrate a strong correlation between individually assayed switches and high-throughput assayed switches, and also show that the number of sorting bins used to produce our high-throughput toehold switch dataset was optimal. Guided by the reviewers' comments and suggestions, we have also adjusted the language in the main text and supplementary information accordingly. Below we include a point-by-point response to all of the reviewers' comments.

Reviewer #1 (Remarks to the Author):

The paper presents a deep learning approach to resolve structure-function relationships in the context of toehold switches. In order to train a model that maps a toehold sequence to its ON/OFF foldchange the group generated around 100 000 toehold sequences of length 148. From the 148 bases roughly half of them remain fixed in order to deviate not too far from the canonical toehold structure.

First of all, generating and characterizing such a library is in itself a huge endeavor. If released to the public, the dataset would be very valuable for the scientific community for follow-up studies etc.

The trained neural network where the sequence is directly entered as a one-hot encoding shows a dramatic improvement in the predictive power over a network that only receives thermodynamic and kinetic features of the sequence or over purely biophysical models.

Apart from that demonstration, the work makes some effort to trace back the biophysical basis of why certain toehold sequences score high. In particular, the second input encoding in terms of the complementarity map is well motivated and allows us to learn something about the secondary structure features by computing saliency maps.

I have some remarks and questions that the authors should address:

Reviewer 1 Comment 1:

I have some concerns regarding the data normalization as given in the M&M section.

It appears as if authors are normalizing the ON and OFF flow-seq data separately to their maximum (see Fig. 2 A & B). This would remove the actual foldchange and would only allow to rank structures relative to each other somehow. It would be not of a concern if both flow datasets are normalized to one common number. From flow-data in Fig S2 one finds (surprisingly) that the maximum of both, the ON and OFF library are roughly the same, which would indicate that the apparently performed separate normalization would not generate too much of a problem. However, simple calculus then says that the computed fold change is simply the log ratio of the raw data in the linear domain scaled by some constant (which might be easier to communicate). I do not believe that separate normalization is a good thing to do and the recurring theme in the paper that the biophysical models to do not perform well and do better on Spearman correlations, might relate to this normalization issue. Authors should make that exposition of data processing more precise. Perhaps it is worthwhile to plot the raw flow-seq data where the not normalized ON and OFF distributions are shown on one scale.

Author Response:

We appreciate the reviewer's concerns on how the data were normalized, for which we have further clarified our normalization procedure in the revised text. It appears that due to unclear language in our manuscript, we were previously unable to convey the exact methodology used to normalize the collected ON and OFF signals. While the reviewer is correct in that the two states, ON and OFF, were normalized to an arbitrary number range, they were actually normalized on the same scale, namely, the four sorting bins used to sort both libraries that were held constant. The ON and OFF libraries were both sorted on the same instrument, one after the other on the same day under identical induction conditions, using fixed settings on the instrument bounded by a positive control high-performing switch for the highest bin (corresponding to GFP signal 1) and a negative control empty pUC19 vector for the lowest bin (corresponding to GFP signal 0). Because the four sorting bins used were the same for both the ON and OFF libraries, the minimum and maximum GFP signal bounded by the upper and lower bins used to normalize the data from 0 to 1 are the same for both ON and OFF, and hence the normalized values in the ON and OFF datasets are directly comparable. As the reviewer notes, "It would be not of a concern if both flow datasets are normalized to one common number," and this is in fact the case. This is now presented more clearly in our manuscript with the following altered explanations in the Methods section "Flow-Seq Pipeline":

"To facilitate comparison between the ON and OFF libraries and to ensure both would be measured on the same scale of GFP signal, we utilized two control plasmids to anchor the lowest and highest GFP expression levels for sorting. A high-performing switch from Green et al. (1), referred to by the authors as Switch #4, was cloned both in its OFF state and in the modified, fused-trigger ON state. The Switch #4 ON state expressed at very high levels in our assay and, when compared to the full library distributions of all ON and OFF variants, this control switch marked the highest total levels of GFP signal (Fig. S2). We thus used the Switch #4 distribution to demarcate the highest bin of activity. We used the pUC19 plasmid as a negative control to mark the lowest bin of GFP signal (Fig. S2A), since it does not contain GFP."

This is also now presented more clearly in the Methods section “Deep Sequencing, Read Data Processing and Read Count Analysis”:

“Frequencies of each variant were tabulated for each cell-sorted bin and normalized to the total reads per bin. Each variant’s functional value was computed as the weighted mean of its normalized frequencies across all bins. Because each library was sorted using the same gates established by the control plasmids (see Methods for “Flow-seq Pipeline”), and since each library spanned a remarkably similar range of minimum and maximum GFP intensity (Fig. S2A,B), we scaled the ON and OFF values for each variant to fall between [0,1]. A value of 0 was given to a variant if all corresponding reads were found only in the lowest bin and a value of 1 if all corresponding reads were found only in the highest bin. An ON/OFF metric was calculated by subtracting these individual ON and OFF signal metrics (Fig. 1E), which resulted in values between [-1,1].”

We have also added panels to Fig. S2 showing the scaled dataset values converted from [0,1] back to their original raw log-scale GFP fluorescence, as per the reviewer’s suggestion.

The reviewer also points out the interesting fact that the maximum and minimum GFP signals in both the OFF and ON state libraries are the same, which is in fact the case. Many faulty OFF-state switches with high degrees of leakage yielded fully ON-like states with a maximum GFP intensity, and many faulty ON-state switches were fully OFF with no detectable GFP intensity. These findings can be explained by OFF state switches that do not fold correctly and leave the RBS exposed, or ON state switches whose triggers do not efficiently unfold the switch hairpin stem and leave the RBS sequestered. To avoid any confusion surrounding this counterintuitive finding, we have added language to the Methods section “Flow-Seq Pipeline”, specifically pointing it out:

“We found the GFP expression levels for each library for ON and OFF variants contained a full spectrum of activity between the levels of the control plasmids utilized (Fig. S2). For example, faulty OFF-state switches with high degrees of leaky GFP expression yielded fully ON-like states with maximum GFP intensity, likely because incorrect folding resulted in leaving the RBS exposed. Similarly, faulty ON-state switches had the lowest, negative control levels of GFP intensity, presumably because those variants’ triggers could not efficiently unfold the switch hairpin stem thus leaving the RBS sequestered. “

We have also added the following language to the main text section “Library synthesis, characterization, and validation”:

“Both ON and OFF data spanned the full range of measured GFP signals, meaning that some ON switches failed to induce and expressed no measurable GFP signal, while some OFF switches failed to repress ribosome binding and leaked the maximum measurable GFP signal.”

Reviewer 1 Comment 2:

From Fig S2 it is also clear that the dataset is very unbalanced. For example, taking the ON dataset, the regression error will be dominated by structures that actually have a low ON level; same is true (and more severe) for the classification task. Hence, the network is provided with very few data points to learn switch features corresponding to high ON levels. That is consistent with Fig. S9, where one sees that the network (even the one-hot) poorly predicts high fold-change switches (more or less none is exceeding a ON/OFF ratio of 0.6 while experimental data goes up to one. The authors should state more clearly how they approached the problem that the dataset was unbalanced and be more explicit in the mentioned limitations.

Author Response:

We thank the reviewer for highlighting important concerns regarding the distribution of the functional data and improvements that can be made when modeling such unbalanced data. The reviewer correctly points out that the distribution of GFP signal in the flow-sorting data displayed in Figure S2 is highly unbalanced for both the ON and OFF libraries. A large fraction of this experimental unbalancing was determined by high-throughput sequencing to be due to low-signal cells that contained incorrectly synthesized oligomers with one of the following: truncations lacking a start codon, truncations lacking an SD sequence, or variants containing a frameshift. Error-containing constructs were thus disproportionately high in the lowest GFP sorting bin, so we divided the read counts for each correct design by the total for all designs for each bin. We have adjusted the language in the Methods section “Deep Sequencing, Read Data Processing and Read Count Analysis” to clarify this technical consideration:

“The distribution of GFP signal in the flow-sorting data displayed in Fig. S2A is highly imbalanced for both the ON and OFF libraries. A large fraction of the oligo library pool contained incorrectly synthesized oligomers. These were largely truncated products lacking a start codon, lacking an SD sequence, or containing a frameshift that we would expect to lead to low GFP signal. We estimate that at least 50% of the cells that we sorted contained such a truncated variant, and most of these ended up in the lowest bin. Thus, only sequences matching our intended designs were retained for further analysis. For the ON and OFF libraries, respectively, 10,390,207 reads and 20,788,966 reads were mapped to a correct switch sequence. The final ON and OFF libraries seen in Fig. 2 are notably less skewed than the flow-sorting data seen in Fig. S2 thanks to the exclusion of reads corresponding to incorrectly synthesized switches.”

As a result, the toehold switch dataset itself is not as imbalanced as the raw fluorescence measurements obtained during sorting, because the toehold switch dataset excludes the relatively large fraction of constructs corresponding to incorrectly synthesized constructs, most of which are truncation variants falling in the lowest bin. The ON state library in particular is quite close to being uniformly distributed.

However, the reviewer is correct that the OFF state library remains unbalanced, even after accounting for the removal of truncated low-signal constructs during data analysis.

We now realize this was not clearly stated in the main text of our manuscript, and in order to bring attention to it as a potential limitation we have added the following language to the main text section “Library synthesis, characterization, and validation”:

“Additionally, it should be noted that while ON data are relatively uniform in distribution, OFF data are highly skewed towards low-signal variants (see Supplementary methods section for a detailed discussion of data balancing).”

To follow up on the reviewer’s concern, we also performed new experiments evaluating the performance of our MLP models on balanced categorical and continuous data, using four common approaches: (1) rank-order transformation of continuous data to a uniform distribution, and (2) re-sampling of under-represented continuous data points to achieve a balanced distribution, (3) removal of excess categorical classes, and (4) duplication of under-represented categorical classes. The new data are presented in Fig. S14 and Fig. S15. Interestingly, none of these efforts to create a less skewed dataset improved accuracy, suggesting that our models were already optimally compensating for this limitation. The following detailed discussion of these considerations has been added to the new Methods section “Data Balancing”:

“As part of a wide-reaching parameter search performed while optimizing our deep learning models, we attempted four approaches to address the limitation of skewed OFF state data (enumerated below). Interestingly, we found that these only gave at most very small improvements in model accuracy as measured by R^2 , AUROC, or AUPRC (Fig. S14, S15). This suggested to us that by using un-transformed and unbalanced data our models were already achieving nearly the best performance possible with those architectures. A trade-off of using unbalanced data is predictions often center around the total mean of the distribution. We utilized a variety of performance metrics, especially the AUPRC, to aid interpretation of modeling unbalanced data. To compare the performance of various balancing strategies, we performed the following:

- 1. During regression, we transformed ON, OFF, and ONOFF data into a uniform distribution using sklearn QuantileTransform before training the model, and then transformed predicted test set data back to their original values to calculate accuracy metrics. This transformation retained the rank-order of the data.*
- 2. During regression, we balanced ON, OFF, and ONOFF data into a uniform distribution by splitting the data into twenty bins and randomly re-sampling data points from under-represented bins, done only for training and validation data. For withheld testing data, data points were randomly removed from over-represented bins until a uniform distribution was achieved in order to show predictive performance across the range of datapoints.*
- 3. During binary classification of ON/OFF, we balanced the high and low classes by randomly removing entries from the over-represented lower class until the two classes contained the same number of entries.*
- 4. During binary classification of ON/OFF, we balanced the high and low classes by randomly duplicating entries from the under-represented higher class until the two classes contained the same number of entries.*

One factor that affected model accuracy was the cutoff for binary classification of ON/OFF. Increasing the cutoff for the high and low classes changed how imbalanced the ON/OFF data was, and had a significant effect on both AUROC and AUPRC. We carefully analyzed the implications of this technical consideration and described the decision we made to place the cutoff at ON/OFF=0.7 (classifying for the top 8.3% of ON/OFF values) in Fig. S8.”

Reviewer 1 Comment 3:

Staying with S9, one sees clear artifacts of the performed Flow-seq pipeline, where the pronounced grid lines are apparently due to those sequences that were exclusively detected in one FACS bin. Those accumulation points are obviously not physics but artifacts (such artifacts only start to disappear at QC5 according to Fig S13). Although authors discuss different QCs, I would like to see a more thorough and transparent discussion about those things and how they may affect the overall conclusions – in particular those that the biophysical models are performing so poorly (in contrast to deep nets, biophysics cannot fit artifacts).

Author Response:

We thank the reviewer for requesting a more thorough discussion of the binning and QC approaches. We agree with the observation that deep neural network models might perform better in fitting such artifacts and that a simple comparison at the lowest QC levels would not be fair or appropriate. We had this in mind when developing the various QC levels, and have added a new Fig. S16 carefully comparing the correlation of existing biophysical parameters against all datasets at quality control levels above QC2, and have also added new data to Fig. S4 comparing the performance of a neural network trained on those biophysical parameters at higher quality control levels (with QC5 being the most stringent set that did not contain any sorting artifacts). Importantly, we did not observe any correlation values between our data and the biophysical rational parameters that meaningfully differed at higher QC levels compared with the data presented in Fig. 3 (which was at QC2), and in fact many correlations were lower at higher QC levels. A notable exception is the correlation between the Salis Lab RBS Calculator OFF prediction and the ON/OFF measurement, which increases from 0.011 in QC3 to 0.061 in QC5. However, an R^2 accuracy of 0.061 is still reasonably negligible, and consequently is not relevant enough to change the overall conclusions of these results. Given this information, it seems unlikely that the failure of the biophysical rational parameters to predict our data was due to sorting-based artifacts in the dataset. Nonetheless, we highlight these important considerations with additional language in the Methods section “Library Quality Control” in order to improve the transparency and thoroughness of our descriptions of model comparisons:

“Artifacts of the flow-seq pipeline are also clearly visible in lower-QC datasets (see Fig. S13). These manifest as “spikes” of intensity at the borders of the sorting gates, corresponding to an overrepresentation of variants with reads in only one bin. As read count thresholds increase and sampling improves, such variants become rarer – the

artifacts are visible in QC1 and QC2, less visible in QC3 and QC4, and largely absent in QC5. Given the possibility that models trained on different data inputs (sequence-only vs biophysical) might fit data with such local distortions to differing degrees, we also analyzed the performance of existing biophysical models and neural network models trained on biophysical parameters against all datasets at QC levels above QC2, with QC5 being the most stringent set that did not contain any apparent sorting artifacts (Fig. S4). We did not observe a meaningful improvement in R^2 accuracy using an MLP trained on the biophysical rational parameters at QC levels 1-4 and then tested on QC5. Neither were significant improvements in R^2 correlation seen between the data and individual biophysical parameters at higher QC levels (Fig. S16). We are therefore confident that at the quality control level chosen for the final dataset (QC2), the sorting artifacts did not differentially impact model performance.”

Reviewer 1 Comment 4:

Complexity and data-requirements for training strongly depend on the input dimension of the problem. Hence, in the traditional one-hot encoding I do not see any point in inputting also sequence positions that remain unaltered through the entire dataset. A network of reduced input dimension would train faster and would generalize better and would require less training data. Even for the complementarity map input one can envision such a reduced input representation. The author may do the corresponding experiment for the supplement.

Author Response:

We thank the reviewer for bringing to attention the fact that a reduced input representation is possible since portions of the sequence are held constant. As the reviewer notes, the use of a smaller input representation would train faster. However, we did not approach any limitations on training time for the already short sequences (<150 bases) containing the constant regions. We chose to include the full sequence due to the possibility of secondary structure interactions between variable regions and constant regions, which we had hoped certain models would capture, particularly higher-order models such as the CNN and the LSTM. These sequence interactions were especially important to model explicitly for the VIS4Map application, in order to provide a physically interpretable visual complementarity map representation, since eliminating the constant regions from the input would obscure interactions between the constant and variable regions (for example, putative GFP linker–stem interactions observed in Fig. S12). In order to address the reviewer’s concern, we performed a comparison between an MLP model trained on a one-hot representation of either the entire toehold switch sequence, or the trigger sequence only. We found that very slight but statistically significant differences were observable, and have included this data figure in a new Fig. S14C, as per the reviewer’s recommendation.

Reviewer 1 Comment 5:

As ON and OFF level are also normalized and hence ON/OFF level follow a simple deterministic mapping, I do not see the point of trying to regress ON, OFF and ON/OFF

values with the network. ON/OFF values can just be computed from the two ON and OFF network outputs.

Author Response:

The reviewer is correct in pointing out that the ON/OFF values can be computed directly from the prediction of ON and OFF values, separately. However, the ON/OFF value will be a more practically useful metric for many readers, given that it is the ultimate performance metric desired for most applications of toehold switches. Many readers will want to know how well the various models perform in predicting both ON and OFF at the same time, which is more difficult than predicting them separately (hence why ON/OFF accuracy is consistently lower than ON or OFF accuracy). While one can deterministically calculate the ON/OFF value of any individual switch from its ON and OFF, it would not be possible (or at least be difficult) for a reader to calculate the R^2 accuracy of a model in predicting ON/OFF based on its R^2 in predicting ON and OFF separately. Hence, we respectfully argue that it actually benefits the paper to report the performance of our tested models in regressing ON, OFF, and also ON/OFF as shown in Fig. 2, especially for readers interested in the practical implementation of our methods.

Reviewer 1 Comment 6:

The references appear incomplete, i.e. there has been earlier work on optimizing riboswitches using machine learning (Groher A-C, et al. ACS Synthetic Biology, 2018). As for the presented comparison of cell-free versus in-vivo for riboregulators, there has been work supporting the made point that the rank order of switches is preserved but generally not their fold-change (e.g. Lehr F-X et al. ACS Synthetic Biology, 2019).

Author Response:

We thank the reviewer for alerting us to these associated contributions and we have added the following references to the revised manuscript accordingly:

22. Groher, Ann-Christin, et al. "Tuning the Performance of Synthetic Riboswitches Using Machine Learning." ACS Synthetic Biology, vol. 8, no. 1, Apr. 2018, pp. 34–44., doi:10.1021/acssynbio.8b00207.
24. Peterman, Neil, and Erel Levine. "Sort-Seq under the Hood: Implications of Design Choices on Large-Scale Characterization of Sequence-Function Relations." BMC Genomics, vol. 17, no. 1, Sept. 2016, doi:10.1186/s12864-016-2533-5.
25. Cambray, Guillaume, et al. "Evaluation of 244,000 Synthetic Sequences Reveals Design Principles to Optimize Translation in Escherichia Coli." *Nature Biotechnology*, vol. 36, no. 10, 2018, pp. 1005–1015., doi:10.1038/nbt.4238.
26. Kinney, J., McCandlish, D. (2019). Massively Parallel Assays and Quantitative Sequence-Function Relationships. *Annual Review of Genomics and Human Genetics*. 20(1), 99-127. doi: 10.1146/annurev-genom-083118-014845
27. Kinney, J., Murugan, A., Callan, C., Cox, E. (2010). Using deep sequencing to characterize the biophysical mechanism of a transcriptional regulatory sequence/

- Proceedings of the National Academy of Sciences*. 107(20), 9158-9163. doi: 10.1073/pnas.1004290107
28. François-Xavier Lehr, Maleen Hanst, Marc Vogel, Jennifer Kremer, H. Ulrich Göringer, Beatrix Suess, and Heinz Koepl Cell-Free Prototyping of AND-Logic Gates Based on Heterogeneous RNA Activators *ACS Synthetic Biology* 2019 8 (9), 2163-2173

Regarding the comparison we present between cell-free vs *in-vivo* toehold switches, the reviewer is right to point to the apparent lack of correlation between them in Fig. 3D. While it is certainly true that in many contexts the rank order of riboswitch performance can be directly compared between *in vivo* and *in vitro* environments, in our own hands and with the particular toehold switch architecture selected for this dataset, the correlation has been relatively weaker. One possible explanation might involve differences in the hybridization of short RNA strands (with potentially low T_m as our toehold is 12nt long) in the cellular environment which experiences significant molecular crowding versus more dilute environments like the PURExpress system. In order to demonstrate this, we have performed additional experiments measuring the cell-free induction signal observed from twenty previously published *in vivo* switches, now present in Fig. S1C. Additionally, in order to make clearer to the reader differences between *in vitro* and *in vivo* use of this particular toehold switch, we have altered our language describing said relationship in the main text:

*“All low-performance switches showed no induction, while the high-performance switches showed a spread of ON/OFF ratios between 2 and 10 ($p < 0.0001$ between high and low switches, two-tailed t -test). The wide range of GFP expression seen from the high-performance switches results from a relatively weak rank-order correlation we have observed between the performance of our toeholds *in vivo* and *in vitro* (Fig. S1C), which differs from other work comparing RNA actuators in living cells and cell-free systems (28). The effect may stem from differences in trigger-toehold interactions between the *in vivo* cellular environment and the *in-vitro* cell-free environment. Nonetheless, these results indicate that while the performance of toehold switches *in vivo* and *in vitro* may differ, *in vivo* measurements can still be used to classify categorically whether a switch will function *in vitro*.”*

Reviewer 1 Comment 7:

It remains unclear to me, why authors did not take the next logical step of using the deep network as an inverse model to find novel toeholds with a high fold-change. This can either be done by sampling in combination with active learning / Bayesian optimization as has been done for protein engineering already (e.g. review of Yang et al. *Nature Methods*, 2019) or with some other back-tracking algorithms.

Author Response:

We thank the reviewer for the insight to use the model for novel toehold prediction. For our manuscript, we focused on the application of toeholds for known targets, as opposed to arbitrary targets. The known target space we sampled included the full genomes of

more than a dozen pathogenic viruses and nearly 1,000 coding regions of human transcription factors. The improved modeling and associated predictive power are primarily geared towards the selection of the best target-switch combination when many possible targets can be selected, such as for the detection of a novel virus or new transcript. We therefore have chosen not to use the deep neural network as an inverse model to find novel toeholds with a high fold-change. This reviewer may be interested to know that such an application is a distinguishing focus of a companion manuscript entitled, “Sequence-to-function deep learning frameworks for engineered riboregulators”, that has been co-submitted to *Nature Communications* by a collaborative team led by our colleagues, Timothy Lu (MIT) and Diogo Camacho (Wyss Institute, Harvard).

Reviewer #2 (Remarks to the Author):

This is an interesting paper on modeling RNA switches that can be published if a few minor issues are addressed:

Reviewer 2 Comment 1:

Why were only 4 bins used for the flow-seq experiment? And how quantitative are off-state measurements given background fluorescence? More detail on the sequencing strategy should be included in the main paper.

Author Response:

We appreciate the reviewer’s request for more clarification on the experimental design, including the choice of number of bins for cell sorting and the background level of fluorescence. In the revised manuscript, we have included a more thorough discussion of these issues. Each bin corresponds to log₁₀ increase in fluorescence intensity, such that our sorting scheme bins across four orders of magnitude. We used a weighted averaging approach for a given variant that takes the abundance of the variant in each bin, which produces a continuous value across the range of detection. Our preliminary studies of clonal toehold variants transformed into millions of cells showed a normal distribution of intensity that roughly spanned a full order of magnitude. Additional bins at the lowest and highest end of the detection range would not have logically increased fidelity of those mean intensities. Nonetheless, in order to empirically assess any limitations due to the number of bins used, we have performed additional experiments applying our flow-seq pipeline to a panel of 20 previously characterized switches from Green et al. (2014), using identical induction and sorting conditions as for our main toehold switch dataset. These results are now presented in Fig. S1D,E, and show no significant change in signal when splitting the four previously used bins into eight bins. We have added more detail about the sequencing protocol and strategy in the Methods section “Flow-Seq Pipeline”:

“The number of bins used was chosen based on a preliminary study of our flow-seq pipeline characterizing a panel of 20 previously published switches from Green et al (1) (Fig. S1D). Clonal toehold variants showed a normal distribution of intensity that roughly spanned an order of magnitude (as seen for Switch #4 in Figure S2), and no

difference in measured flow-seq signal was observed when sorting on four or eight bins, suggesting that four bins was sufficient to accurately measure fluorescence across four orders of magnitude in high-throughput (Fig. S1E). Nonetheless it should be noted that at lower read sampling thresholds, artifacts were observable at the borders of the four bins (Figure S13, see “Library Quality Control” for a further discussion of these artifacts).”

Reviewer 2 Comment 2:

The data in Fig. 2D suggest that flow-seq based measurements are not very correlated with in vitro fluorescence measurements. The fluorescence values vary over a 5-fold range even though all “good” switches presumably have similar fold change as measured by sequencing (ON/OFF>0.97). Would it be better to compare flow-seq data to measurements performed with individual switches in vivo (i.e. flow cytometry using just a single switch at a time) and is the relative lack of quantitative agreement due to different switch behavior in cells and in vitro or due to facs-seq noise?

Author Response:

The reviewer is correct that the correlation between our flow-seq measurements and switches tested under cell-free conditions is not as strong as one might expect. However, we do not think this stems from noise in our flow-seq pipeline. The 16 switches tested under cell-free conditions for Figure 2D were taken from the highest-quality portion of our dataset (QC5, with read counts over 300 per switch), and the correlation between flow-seq replicates in that portion of the dataset was quite high ($R^2 \sim 0.8$, Figure S3B), suggesting that the flow-seq assay reliably measured *in-vivo* fluorescence in a way that should at least preserve rank-order correlation. Nonetheless, to address this concern, we have performed additional experiments applying our flow-seq pipeline to a panel of 20 previously characterized switches from Green et al. (2014), using identical induction and sorting conditions as for our main toehold switch dataset. These results are now presented in a new Fig. S1D,E, and show a strong correlation between individually measured fluorescence as previously reported and the pooled measurements of our flow-seq pipeline (Pearson $R=0.788$, Spearman $R=0.842$).

We believe the majority of this noise stems from differences between the behavior of our toehold switches in cells compared with cell-free expression systems. To further illustrate this, we performed additional experiments assaying 20 switches in a cell-free expression system that had previously been assayed *in vivo* by Green et al. (2014). Comparing the *in-vivo* ON state induction from Green et al. to cell-free ON state induction, we can see that while low-signal switches *in vivo* are reliably low in cell-free conditions, many switches that induce well *in vivo* do not produce proportionally as much signal in cell-free conditions. One possible explanation for this behavior is that trigger binding behavior might differ between the two environments. We have included these data in a new Fig. S1C, and have included additional language to the main text section “Library synthesis, characterization, and validation” addressing these concerns:

“All low-performance switches showed no induction, while the high-performance switches showed a spread of ON/OFF ratios between 2 and 10 ($p < 0.0001$ between high

and low switches, two-tailed t-test). The wide range of GFP expression seen from the high-performance switches results from a relatively weak rank-order correlation we have observed between the performance of our toeholds in vivo and in vitro (Fig. S1C), which differs from other work comparing RNA actuators in living cells and cell-free systems (28). The effect may stem from differences in trigger-toehold interactions between the in vivo cellular environment and the in-vitro cell-free environment. Nonetheless, these results indicate that while the performance of toehold switches in vivo and in vitro may differ, in vivo measurements can still be used to classify categorically whether a switch will function in vitro.”

Reviewer 2 Comment 3:

What test set was used for the data shown in fig.3? please add this information. If it was randomly selected, it would be good to also try an alternative test set consisting of the most deeply sequenced switches. Such a test set (which should still be randomly distributed in sequence space) should help reduce noise due to the measurement. Even the best model presented only has $R^2=0.43$ for On/Off predictions and I wonder whether this is an artifact of noisy measurements.

Author Response:

The reviewer is correct that the test sets used during cross validation in Fig. 3 were randomly selected at the sequence depth cutoff of 10 reads or higher for both ON and OFF measurements (the QC2 dataset, see Table S1), and points out that this information is relevant to include in the figure caption and not merely in the Methods section. We have therefore altered the caption of Fig. 3 as follows:

“(D) Box and whisker plots for R^2 between experimental and regression-based predictions for best performing rational features, logistic regression models and MLPs using ten-fold cross validation (test sets randomly selected from quality control process #2, QC2 in Fig. S13 and Table S1). (E) Mean absolute error (MAE) between experimental and predicted values for these same models. (F) Box and whisker plots for area under the curve (AUC) of the receiver-operator curve (ROC) and the precision-recall curve (P-R) in classification-mode predictions compared to experimental values using three-fold cross validation (test sets randomly selected from quality control process #2, QC2 in Fig. S13 and Table S1).”

The reviewer further suggests that a test set consisting of the most deeply sequenced switches be withheld before training, allowing for the model to predict data that is less influenced by measurement noise caused by lower read count sampling. We perform such an analysis in Fig. S4 using an MLP trained either on a one-hot sequence encoding, or on thirty rational thermodynamic features. We withheld the QC5 dataset for testing (which contains variants with read counts of 300 or more and with high/low standard deviation cutoffs, details in Table S1) and trained with datasets of increasingly stringent read depth (QC1, QC2, QC3, QC4). Notably we did not observe increased performance of the MLP on the QC5 test set compared with random test sets from QC2 (the results of Figure 3), nor did we observe improved results with read count depths more stringent than that of

the QC2 dataset. Therefore while the reviewer is correct in wondering whether greater read depth would improve the results of our models, we conclude from these results that the QC2 level of read depth is optimal for balancing dataset size with flow-seq measurement noise.

Reviewer 2 Comment 4:

This statement is outdated: “One significant drawback of using deep learning to predict biological function is the inherent difficulty in understanding learned patterns in a way that helps researchers to elucidate biological mechanisms underlying model predictions.” There is extensive recent work on visualizing DNA and RNA sequence features, mapping filter motifs to biologically and even using such models to guide sequence design for synthetic biology.

Author Response:

We appreciate the reviewer’s correction of the outdated statement and have added several new references supporting the recent work in visualizing sequence features:

45. Kelley DR, Snoek J, Rinn JL. Basset: learning the regulatory code of the accessible genome with deep convolutional neural networks *Genome Res.* 2016 Jul;26(7):990-9. doi: 10.1101/gr.200535.115.
46. Simon Höllerer, Laetitia Papaxanthos, Anja Cathrin Gumpinger, Katrin Fischer, Christian Beisel, Karsten Borgwardt, Yaakov Benenson, Markus Jeschek (2020) Large-scale DNA-based phenotypic recording and deep learning enable highly accurate sequence-function mapping. *bioRxiv* 2020.01.23.915405; doi: <https://doi.org/10.1101/2020.01.23.915405>
50. Singh, J., Hanson, J., Paliwal, K. et al. RNA secondary structure prediction using an ensemble of two-dimensional deep neural networks and transfer learning. *Nat Commun* 10, 5407 (2019).

We have also added language mentioning the recent work and describing how our secondary structure method complements and extends the work in the field in the main text section “Visualizing learned RNA secondary structure motifs with VIS4Map”:

“Recent work has been developed to visualize sequence features by mapping learned convolutional filters to biologically-relevant sequence motifs (45, 46). Additional methods have been established to address how models link biological theory, including alternative network architectures (47), and the use of saliency maps (48, 49), which reveal the regions of an input that deep learning models weigh most heavily and therefore pay the most attention to when making predictions. While saliency maps have been previously used to visualize model attention in one-hot representations of sequence data (10, 17, 18, 20, 48), such implementations focus only on the primary sequence and have not been developed to identify salient secondary structure interactions, which are especially relevant in the operation of RNA synthetic biology elements. Furthermore, prior work related to RNA secondary structure prediction using deep learning (50) has not utilized saliency techniques to highlight relevant secondary structure regions that

lead to improved function in RNA sensors. Instead, visualized representations have been constrained to predetermined structures based on the predictions of thermodynamic models (43, 44), whose abstractions we have found cause significant information loss.”

Reviewer 2 Comment 5:

The work shown in Fig. 5 is very interesting. The authors should use the model to explicitly predict on/off ratio for their switch test set and show that inclusion of secondary structure gives better results than sequence only models.

Author Response:

We appreciate the reviewer’s comments in this regard. As seen in Fig. 4, we conducted a thorough comparison of the regression and classification accuracies exhibited by our models including a three-layer MLP, a CNN trained on a one-hot sequence input, an LSTM trained on a one-hot sequence input, as well as the model seen in Fig. 5, which corresponds to a CNN with a 2D one-hot complementarity map input.

Upon evaluating both the R^2 and MAE of predictions for ON, OFF, and ON/OFF in regression mode, and the AUROC and AUPRC of predictions for ON/OFF in classification mode for these models (Fig. 4), we concluded that while the 2D-input CNN does provide more insight regarding the relevance of secondary structure for predicting function, this specific neural network architecture did not achieve higher predictive accuracy as compared to the sequence-based, three-layer MLP. This appears to be caused by an increased model capacity, which requires larger dataset sizes to provide the same level of training. To ensure that the reader does not misunderstand this, we have altered the following language in the main text section “Visualizing learned RNA secondary structure motifs with VIS4Map”:

“Encouraged by our CNN’s ability to elucidate putative RNA secondary structure features directly from in silico-generated training data, we applied VIS4Map to our entire experimental toehold switch function dataset. When trained on a complementarity map representation both in regression mode and classification mode (Fig. 5D), VIS4Map significantly outperformed an MLP trained on rational thermodynamic features. However, VIS4Map did not significantly outperform an MLP trained on a one-hot sequence input (as was the case when predicting NUPACK MFE). The failure of VIS4Map to improve predictions compared with a simpler three-layer MLP model likely results from over- or under-fitting of the higher-dimensional input, similar to the case of our other higher capacity models (Fig. 4A,B,C,D).”

Reviewer 2 Comment 6:

I’m a bit confused by the following statements: “We sought to visualize RNA secondary structures learned by our neural networks in a manner unconstrained by thermodynamic modeling.” Which is followed one paragraph later by “we first pre-trained a CNN to predict NUPACK MFE values from complementarity map representations of a randomly selected in silico RNA sequence dataset.” Aren’t these statements contradictory?

Author Response:

We agree with the reviewer that the selected statements are unclear in succession. We have adjusted to text around each statement to clarify the meaning in the context of each statement. The first statement concerns the use of our modeling approach to ascertain secondary structure elements associated with functions that are based on a complementarity map representation, as opposed to traditional thermodynamic structure prediction. The second statement is primarily intended to validate whether the new approach recapitulates the predictions of the traditional approach. We have adjusted the wording and organization of these statements to provide the reader with additional context and to clarify the motivation of these separate points:

“Importantly, because the complementarity map is unconstrained by a priori hypotheses of RNA folding (similarly to our sequence-based MLP models), we anticipated this approach to be able to identify secondary structures that might be overlooked by commonly used thermodynamic and kinetic algorithms, such as NUPACK and Kinfold.

To first validate whether our visualization approach could capture any meaningful RNA structure features, we trained a CNN to predict NUPACK MFE values from a complementarity map representation of a randomly selected in silico RNA sequence dataset. Because NUPACK’s calculated MFE is directly determined by a predicted RNA secondary structure, we anticipated that a CNN undergoing this training would likely pay attention to secondary structure features, a situation that was confirmed through visualization of individual attention maps (Fig. 5B,C).”

Reviewer 2 Comment 7:

The title is too general. Toehold switches are interesting but constitute only a very specific subset of RNA synthetic biology.

Author Response:

We agree with the reviewer that a more appropriate title better captures the contributions made in the manuscript and have adjusted the title accordingly to: “A deep learning approach programmable RNA switches”.

Reviewers' Comments:

Reviewer #1:

Remarks to the Author:

The authors responded to all my questions and concerns. Where necessary, they performed additional analyses and adapted the MS accordingly. I am happy with the paper and recommend acceptance.

Reviewer #2:

None